# Assembly and the gating mechanism of the Pel exopolysaccharide export complex PelBC of Pseudomonas aeruginosa

Marius Benedens [1,5], Cristian Rosales-Hernandez[2,5], Sabine A. P. Straathof[3], Jennifer Loschwitz [1], Otto Berninghausen [2], Giovanni Maglia [3], Roland Beckmann [2] ✉ & Alexej Kedrov [1,4] ✉

The pathogen *Pseudomonas aeruginosa* enhances its virulence and antibiotic resistance upon formation of durable biofilms. The exopolysaccharides Pel, Psl and alginate essentially contribute to the biofilm matrix, but their secretion mechanisms are barely understood. Here, we reveal the architecture of the outer membrane complex PelBC for Pel export, where the essential periplasmic ring of twelve lipoproteins PelC is mounted on top of the nanodisc-embedded β-barrel PelB. The PelC assembly is stabilized by electrostatic contacts with the periplasmic rim of PelB and via the membrane-anchored acyl chains. The negatively charged interior of the PelB β-barrel forms a route for the cationic Pel exopolysaccharide. The β-barrel is sealed at the extracellular side, but molecular dynamic simulations suggest that the short loop Plug-S is sufficiently flexible to open a tunnel for the exopolysaccharide transport. This gating model is corroborated by single-channel conductivity measurements, where a deletion of Plug-S renders a constitutively open β-barrel. Our structural and functional analysis offers a comprehensive view on this pathogenicity-relevant complex and suggests the route taken by the exopolysaccharide at the final secretion step.

The Gram-negative bacterium *Pseudomonas aeruginosa* is an opportunistic human pathogen that accounts for nearly 20% of all nosocomial infections, being a major risk factor for immunocompromised patients and those with cystic fibrosis[1]. *P. aeruginosa* infections build up a burden for the healthcare systems worldwide due to the prolonged hospitalization period and the associated costs. The primary challenges to combat this versatile bacterium are its rapidly developing antibiotic resistance and formation of extensive durable biofilms, both in tissues upon host invasion and on diverse abiotic surfaces. Stability of the biofilms is mediated by the complex composition of the matrix, where the exopolysaccharides—alginate, Psl and Pel—serve to embed cells, ensure the mechanical strength of the matrix and provide attachment sites for secreted virulence factors[2]. Despite the highest biomedical relevance of the exopolysaccharides of *P. aeruginosa*, the understanding of the molecular mechanisms behind their secretion is poor.

Pel is the major constituent of *P. aeruginosa* pellicles, i.e., films formed at the water-air interface, and it plays a key role at early stages of the biofilm formation at solid surfaces. Pel has been recently identified in other β-, δ-, and γ-proteobacteria, but also several extremophiles and Gram-positive bacteria[3]. *P. aeruginosa* Pel is composed of α −1-4-linked N-acetyl-D-galactosamine residues, which are partially deacetylated in the periplasm, so the Pel exopolysaccharide acquires positive charge along the secretion pathway[4]. This charge is utilized

[1]Synthetic Membrane Systems, Institute of Biochemistry, Heinrich Heine University Düsseldorf, Düsseldorf, Germany. [2]Gene Center Munich, Ludwig Maximilian University of Munich, Munich, Germany. [3]Chemical Biology, Groningen Biomolecular Sciences & Biotechnology Institute, University of Groningen, Groningen, The Netherlands. [4]Interfaculty Center for Membrane Research, Heinrich Heine University Düsseldorf, Düsseldorf, Germany. [5]These authors contributed equally: Marius Benedens, Cristian Rosales-Hernandez. ✉e-mail: beckmann@genzentrum.lmu.de; kedrov@hhu.de

then to crosslink the extracellular DNA thus contributing to the matrix assembly[5]. Other roles of Pel involve stabilization of the adhesin CdrA and complementation for defects in the outer membrane, as deletion of the major structural protein OprF stimulates Pel accumulation at the cell surface (reviewed in ref. 2).

Secretion of Pel in the Gram-negative *P. aeruginosa* assumes the polysaccharide translocation across the inner and outer membranes (Fig. 1), and the genes essential for the process are encoded in a single *pelABCDEFG* operon. The polysaccharide elongation mediated by the glycosyltransferase PelF in the cytoplasm is coupled to the synthase-dependent translocation through the inner membrane complex of PelD, PelE, and PelG subunits, in a still unknown fashion (Fig. 1A)[6]. Upon crossing the periplasm, Pel is partially deacetylated by PelA and then translocated through the outer membrane via PelB[7]. PelB is composed of a C-terminal β-barrel preceded by the helical scaffold, built of multiple tetratricopeptide repeats (TPR). The length of the modelled scaffold exceeds 20 nm, thus being sufficiently long to span the periplasm (Supplementary Fig. 1A). Notably, the N-terminal end of PelB contains 22 apolar and aromatic residues, which may form a transmembrane helix within the inner membrane without a cleavage site for the signal peptidase (Supplementary Fig. 1B). Thus, PelB of *P.*

*aeruginosa*, as well as its multiple homologs in other species, may physically bridge two membranes, with the TPR scaffold forming a passage for the polysaccharide across the periplasm.

The β-barrel of PelB is assumed to form a complex with PelC lipoproteins, a hallmark of the Pel system in Gram-negative bacteria. PelC is essential for the polysaccharide transport, and several mutations within PelC were identified which inhibited the biofilm formation, either via direct interactions with the polysaccharide or due to their involvement in the PelBC complex assembly[8]. The exact role of PelC in the exopolysaccharide translocation is not clear. A partial crystal structure of PelC from *Paraburkholderia phytofirmans* solved in the absence of the membrane and the lipid anchors revealed the protomers arranged into a dodecameric ring with a 3.2 nm-wide pore in the center[8], and the negative charge on the periplasm-facing side of PelC was suggested to assist in translocation of the cationic polysaccharide. The width of the PelC pore matches closely the diameter of the modelled PelB structure (Supplementary Fig. 2), and AlphaFold-based modelling of the complex in 1:12 stoichiometry provides a high-confidence assembly, where the TPR domain of PelB is placed within the central pore of the PelC ring[9]. However, models of a similar confidence could also be rendered at different stoichiometries, either

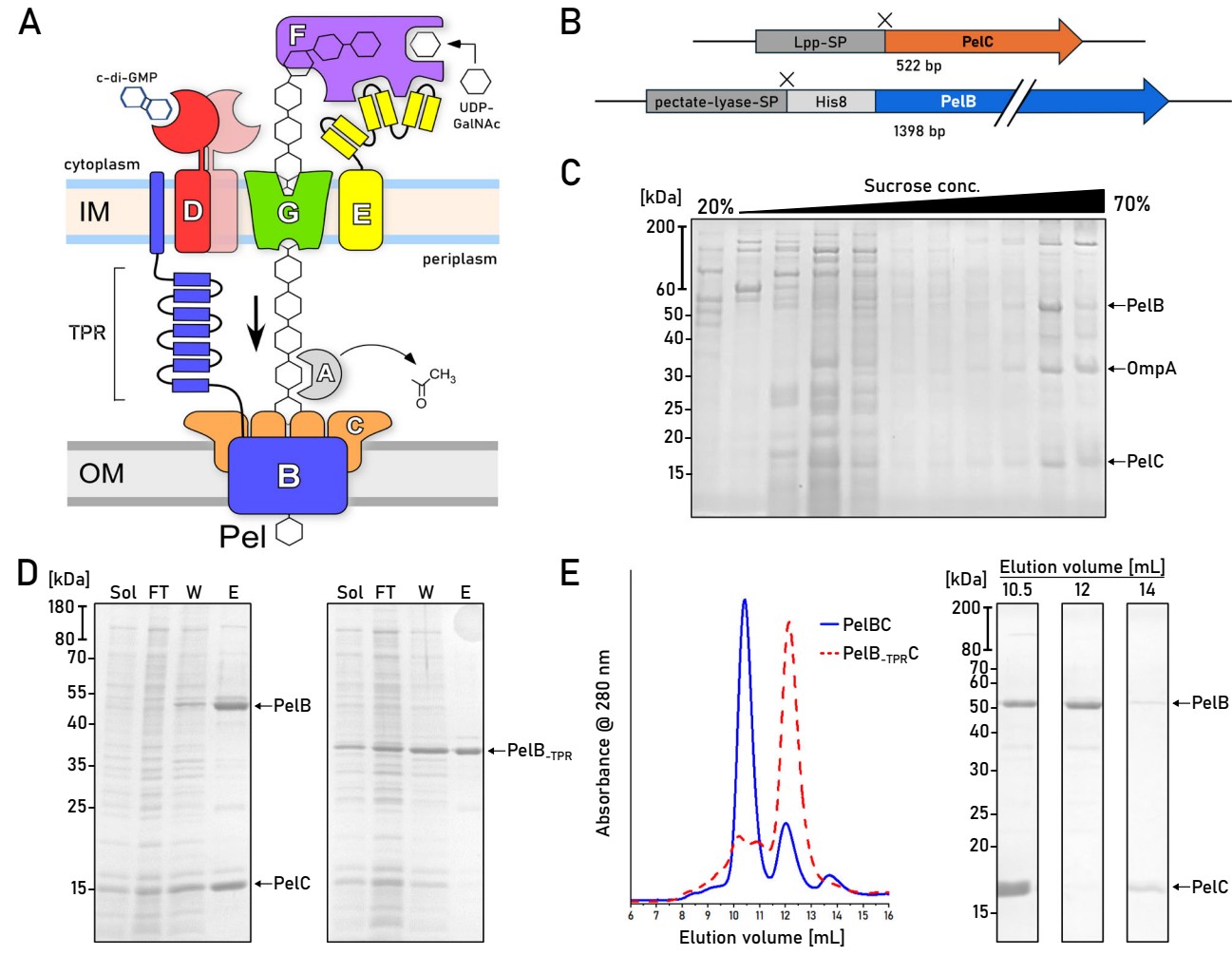

**Fig. 1 | Isolation of the natively assembled PelBC complex of *P. aeruginosa*.**
**A** Putative organization of the synthesis/secretion machinery for the Pel exopoly-saccharide in *P. aeruginosa*. **B** Gene constructs of PelB and PelC in the pETDuet-1 vector. Cleavage sites after the signal peptides (SP) of both proteins are indicated as "X". **C** SDS-PAGE of the membrane isolates after the sucrose gradient shows PelB and PelC co-occurring in late fractions characteristic for the outer membrane vesicles. A reference band for the outer membrane protein A (OmpA) is indicated.

**D** SDS-PAGE of immobilized metal affinity chromatography on PelBC with His-tagged PelB variants, wild-type (left) and the truncated PelB lacking the periplasmic helical scaffold (right, PelB_-TPR). Loaded fractions: "Sol." - detergent-solubilized material; "FT" - flow-through; "W" - wash, "E" - elution. **E** Left: Size-exclusion chromatograms of the PelBC samples from (**D**). Right: SDS-PAGE of the major SEC fractions collected for the wild-type PelBC. Each biochemical experiment was at least repeated twice independently. Source data are provided as a Source Data file.

having less PelC subunits (ratio 1:11) or more (ratios 1:13 and above) (Supplementary Fig. 2). Since the assemblies of lipoproteins in solution and at the membrane interface may differ[10], visualizing the actual architecture of the PelBC complex remains an open task. The experimentally determined structure would also explain the functionally-defective PelC mutants, and potentially address protein:lipid interactions for the lipoproteins and the transmembrane β-barrel. Critically, the modelled PelB barrel does not manifest a substantially wide conduit at the extracellular side for the polysaccharide translocation, being sealed by several loops. Though the model may reflect the idle state of the channel, the experimental structure determination is required to understand the functional dynamics of the complex.

Here, we employed cryogenic electron microscopy (cryo-EM) to visualize the structure of the PelBC complex of *P. aeruginosa* in the lipid-based nanodiscs, reaching a final resolution of 2.5 Å. Next to the overall architecture of the asymmetric complex of 250 kDa, we report extensive protein:lipid interactions formed by both the lipoproteins and the β-barrel and discover that the periplasm-exposed helical domains of PelB and the highly conserved tryptophan of PelC are essential for the assembly of the complex. The acquired structure suggests the route of the exopolysaccharide across the channel, and it is further used to design single-channel conductivity experiments and computational simulations, which jointly provide first evidence for PelB conformational dynamics in the lipid membrane.

## Results

### Isolation and nanodisc reconstitution of the intact PelBC complex

To establish heterologous expression of the PelBC complex, both genes were cloned into pETDuet-1 vector under individual T7 promotors. To facilitate the efficient export of the synthesized proteins into the periplasm via the Sec machinery, the signal sequence of PelC was substituted with the signal sequence of Lpp, the highly abundant Braun's lipoprotein of *Escherichia coli*. PelB was expressed as the C-terminal fragment (residues 762–1193) containing three periplasmic TPR repeats and the transmembrane β-barrel, which were preceded with the conventional signal sequence of pectate lyase B of *Erwinia carotovora* and the octa-histidine tag (Fig. 1B). Based on the AlphaFold models (Supplementary Fig. 2), we assumed that the truncated periplasmic region of PelB would be sufficient to interact with the PelC subunits. Once expressed in *E. coli* C41(DE3) ΔompF ΔacrAB strain[11], the protein localization to the outer membrane was validated by the ultracentrifugation in the sucrose density gradient: Both proteins appeared within the high-density fraction together with OmpA, an intrinsic marker for the outer membrane vesicles (Fig. 1C)[11].

Both PelB and PelC were extracted from the membranes with the mild non-ionic detergent DDM. IMAC based on the N-terminal histidine-tag of PelB resulted in co-purification of substantial amounts of PelC (Fig. 1D), suggesting that the proteins resided as a complex. The PelB-PelC isolates manifested three distinct peaks in size exclusion chromatography (SEC, Fig. 1E): The major peak contained both PelB and the excess of PelC and it was observed at 10.5 mL elution volume suggesting the molecular weight of ~300 kDa, while the downstream peaks at 12 mL and 14 mL contained PelB and PelC, respectively. The appearance of the tag-less free PelC subunits in the latter peak indicated partial disassembly of the detergent-solubilized complex after the IMAC stage. To prevent that, SEC-purified PelBC complexes were immediately reconstituted into nanodiscs in presence of POPC:POPG lipids (molar ratio 70:30). The chosen scaffold protein MSP1D1 builds nanodiscs of ~8 nm inner diameter[12] that should be sufficient to accommodate the β-barrel of PelB and the acyl chain anchors of PelC lipoproteins, assuming the ring-shaped assembly. SEC of PelBC-nanodiscs resulted in a major peak at ~10.5 mL where PelB, PelC, and MSP1D1 were found, followed by minor amounts of PelB-only nanodiscs (Fig. 2A). The lipid environment of nanodiscs greatly stabilized

the complex, as no dissociation of PelC was observed in SEC, so the lipoprotein remained anchored within the lipid layer.

### Cryo-EM resolves the architecture of the PelBC complex in the lipid bilayer

The nanodisc-stabilized PelBC complexes were subjected to cryogenic electron microscopy (cryo-EM). From around 30,000 movies, 4.8 million particles were picked for downstream processing. In reference-free 2D class averages, a balanced distribution of side and top views of the PelBC complex was observed (Fig. 2B and Supplementary Fig. 3), and the nanodisc density could be well-recognized in side views, with the outer diameter of 9 nm in agreement with the previous reports[13]. Notably, no classes displaying stoichiometry different from 1:12 PelB:PelC were obtained. The homogeneity of the particle distribution allowed for a 3D reconstruction of the PelBC complex at 2.5 Å resolution with no symmetry imposed (Supplementary Figs. 3 and 4). Extensive sorting was carried out to look for different conformations of the complex, but no major structural differences between the classes were detected. A focused classification on PelB resulted in a class with the best resolved β-barrel with short α-helical loops and the N-terminal helical extension, which subsequently led to the final map used for model building (Fig. 2C, D). In agreement with the prediction, the complex is composed of a single PelB barrel built of 16 antiparallel β-strands embedded in the lipid bilayer and a dodecameric ring of PelC lipoproteins mounted at the periplasmic side (Fig. 2E). The β-barrel is centrally positioned within the nanodisc, without contacts to the edges which could otherwise distort the conformation. Notably, the architecture of PelB is similar to bacterial transporters for cellulose (BcsC)[14] and PNAG (PgaA)[15], which share the overall fold of the 16-stranded β-barrel, with partially structured loops occluding the extracellular exit of the channel (Supplementary Fig. 6). At the periplasmic side of PelB, two TPR domains resolved from the residues Gly-803 and the intermediate helix ("stalk") are located within the PelC ring, being docked at multiple lipoprotein subunits, as described below. The preceding PelB residues ranging from Asn-762 to Ile-802 remained unresolved, likely due to their higher flexibility.

The interior of the PelB β-barrel is predominantly negatively charged, where the charge is distributed asymmetrically, being largely concentrated on strands 11 to 14 (Fig. 3A). The anionic cluster faces the groove within the TPR domains, so the cationic Pel polysaccharide may employ this route for entering the channel driven by the electrostatic interactions. The anionic wall is continued with the negatively charged and partially structured loops forming the bottom of the barrel at the extracellular side with no tunnel sufficiently wide for the EPS passage (Fig. 3B, C). As a major element here, a short α-helix between the strands 7 and 8 (further referred as Plug-I) is bent inwards the central cavity of PelB where it lays perpendicular to the barrel axis (Fig. 3C). Plug-I forms several contacts within the barrel, first of all via Arg-999 to Tyr-922 in β-strand 2 and Glu-935 in β-strand 3, and so it may serve to stabilize the barrel. Plug-I is opposed by a glycine-rich loop between the C-terminal β-strands 15 and 16 (Plug-S), and together these two loops occlude the exit tunnel of PelB (Fig. 3B). The loop between the β-strands 11 and 12 (Plug-O) is exposed to the solvent at the extracellular side forming a "dome" over the barrel. Though only a small part of this 31 residues-long loop builds an α-helix, the polypeptide chain is well-resolved in cryo-EM, so Plug-O is rigid under the experimental conditions. Comparison of the resolved *P. aeruginosa* PelB structure with the models of PelB homologs from other *Pseudomonas* species shows high conservation of the extracellular loops, apart the Plug-O (Supplementary Figs. 7 and 8). Here, broad variations in the length and the putative structures are observed, suggesting that the loop architecture evolves in response to the certain habitat of the bacteria and/or its specie-specific LPS layer.

Similar to the barrel interiors (Fig. 3), the extracellular side of *P. aeruginosa* PelB is highly anionic with the net charge of −11 (Asp+Glu:

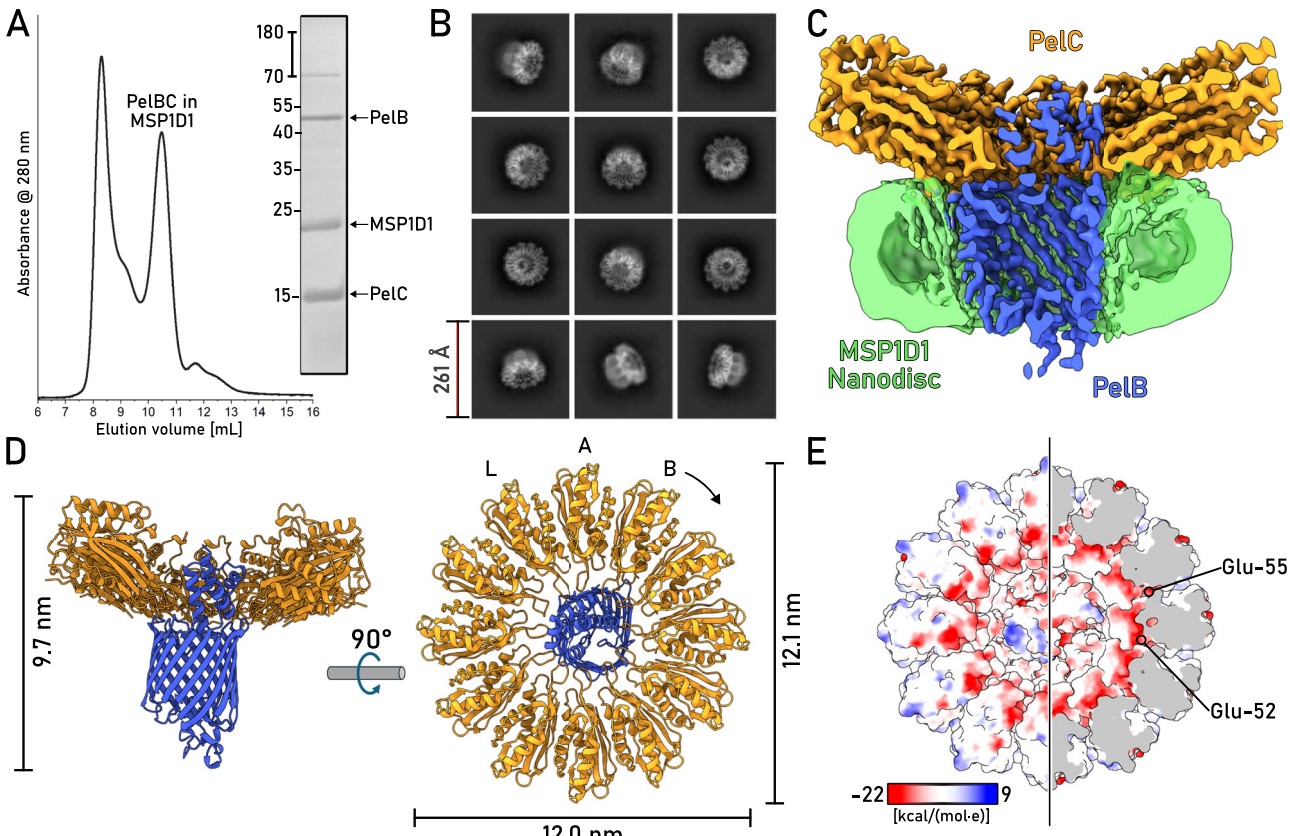

**Fig. 2 | Structure of the PelBC complex embedded into the lipid membrane.**
**A** SEC of PelBC reconstituted into the MSP1D1-based nanodisc and SDS-PAGE of the fraction at 10.5 mL elution volume used for cryo-EM analysis. **B** Characteristic 2D classes found upon cryo-EM data processing show various projections of the assembled PelBC complexes in the nanodiscs. **C** Final 3D reconstruction (cross-section) of the PelBC complex in the nanodisc. PelB is colored in blue, PelC in golden, nanodisc/lipids in green. **D** The structural model of the PelBC complex, PelB in blue, PelC in golden. PelC subunits labeled from "A" to "L", so the subunit A approaches the N-terminal end of the PelB β-barrel. The Plug-S loop occupying the exit tunnel of PelB is seen in the periplasmic view (right). **E** The electrostatic potential at the periplasmic surface of the PelC ring (left) and under the capping loop (right). Each biochemical experiment was at least repeated twice independently. Source data are provided as a Source Data file.

−18, Arg+Lys: 7), which appears as a common feature among PelB homologs in *Pseudomonas* species (Supplementary Fig. 8). Such charge distribution is also seen for several *P. aeruginosa* porins, incl. PA4067 (OprG; PDB ID 2 × 27), PA3280 (OprO; 4RJX), and PA3279 (OprP; 2O4V), and the negative charge may serve to repel the phosphate groups of the surrounding LPS cores. In context of the polysaccharide secretion, it may also be involved in functional opening/closing of the barrel, and/or providing the electrostatic driving force for translocation of the cationic Pel. For the outer membrane transporters of partially cationic polysaccharides, such as bacterial phosphoethanolamine cellulose and PNAG, the electrostatic interactions at the exit of the secretion machinery were proposed to facilitate the directed transport[15].

As a unique feature of the Pel translocation system, twelve PelC subunits assemble in a ring at the periplasmic side of the PelB barrel (Figs. 2 and 4), and so they repeat the architecture of the crystallized PelC of *P. phytofirmans*[8]. Beneficially, the cryo-EM map visualizes the complete periplasm-facing loop-helix turn (residues 70–98) that was not resolved in the crystal structure, due to its flexibility apparent from the current reconstruction (Fig. 4A and Supplementary Fig. 4). The loop-helix segment caps the PelC subunit beneath and shields the excessive negative charge rendered by the glutamates in positions 52 and 55 (Fig. 2E), but also provides negative charge via Asp-79 and Asp-84 lining the central pore (Fig. 4A). At the membrane interface, the extended loop of PelC (further referred as D-loop, for the conserved Asp-119 within) is exposed into the central pore. The electronegative

potential rendered by Asp-119 was previously related to PelC:exopolysaccharide interactions[8], as altering the charge here abolished the biofilm formation. However, within the assembled complex those aspartates are buried at the PelB-PelC interface and build multiple contacts with the surface-exposed arginine residues of PelB, such as 905, 944, 1018, 1114, 1122, and 1190 (Fig. 4B, C), thus playing rather a structural role. Noteworthy, with glycine residues in positions 117 and 120, the D-loop is sufficiently flexible, so individual PelC subunits adapt their conformations to the proximate structural features of PelB, i.e., the helical domain and the periplasmic loops (Fig. 4B).

Two PelC subunits, C and D, build electrostatic interactions with the stalk helix of PelB (Fig. 4D), and the preceding TPRs are stabilized in defined positions by interactions with PelC subunits B/C/-D and I/J/-K, so these two repeats are well-resolved within the central pore. To test the role of the helical region in the PelBC assembly and stability, we co-expressed PelC with the truncated PelB variant containing only the β-barrel domain. Though both proteins could be extracted from the membrane, no PelC was co-purified with PelB in absence of the stalk helix and the TPR domains (Fig. 1D, E), suggesting that the conserved domain is crucial for the protein:protein interaction, and it may serve as a nucleation site for the PelC ring assembly.

## The PelBC complex is stabilized via multiple protein:lipid interactions
The 3D reconstruction of the nanodisc-embedded complex reveals several rod-shaped densities proximate to PelBC (Fig. 5A, B). Within the

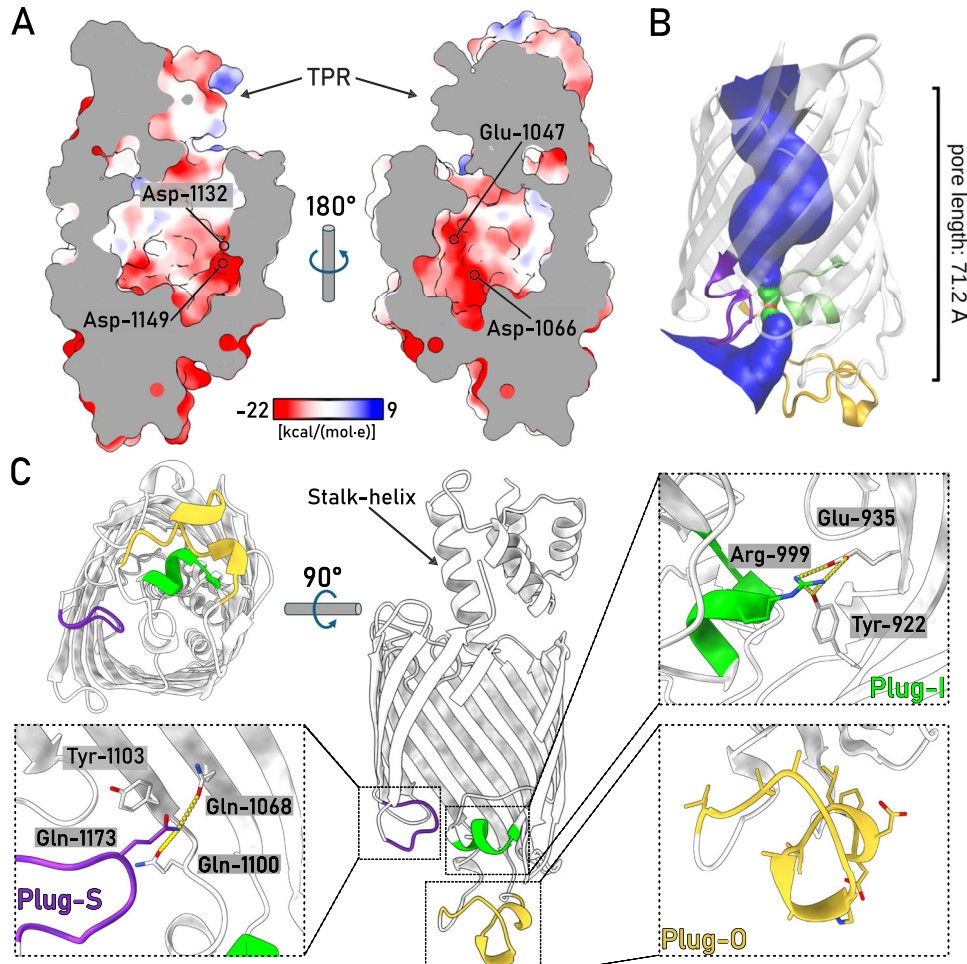

**Fig. 3 | The architecture of the PelB β-barrel. A** The charge distribution inside the PelB β-barrel is asymmetric, with anionic residues localized at the β-strands 11–14 and the bottom of the barrel. Representative residues are indicated. **B** The tunnel across the β-barrel is rendered by HOLE algorithm, shown in blue. The bottleneck impermeable for water molecules is indicated in red, and the areas accessible for a single water molecule are in green. The extracellular loops are colored accordingly to the (**C**). **C** Detailed view on the extracellular loops Plug-I (green), Plug-S (violet), and Plug-O (yellow) of PelB and the interactions which stabilize their positions in the resolved structure. The universally conserved residue Tyr-1103 near the Plug-S loop is indicated.

periplasmic leaflet, most of those densities emerge from the N-termini of PelC subunits, so they are unambiguously assigned to the covalently bound acyl chain anchors. For several PelC subunits, all three acyl chains conjugated to the N-terminal cysteine are resolved, reaching up to 16 carbon atoms (Fig. 5C). The anchor chains of individual PelC subunits are sandwiched between two Trp-149 residues, one from the same PelC, and one from a preceding subunit, a configuration that potentially facilitates docking of the lipoproteins at the membrane interface. Compared to the membrane-less crystal structure[8], Trp-149 residues undergo rotation towards the lipid bilayer (Fig. 4A), so their indole rings immerse into the hydrophobic core of the membrane. Previous in vivo analysis showed that *P. aeruginosa* biofilm formation at the water-air interface was severely inhibited once Trp-149 was replaced by alanine[8]. To test whether this functional defect arose from the complex assembly, we expressed PelC_{W149A} mutant either alone or in combination with PelB. PelC was targeted to the outer membrane in both cases, however we found a detrimental effect of the mutation on stability of the PelBC complex, as only PelB barrel without the lipoproteins could be isolated (Supplementary Fig. 9). Thus, we concluded that Trp-149 stabilizes the lipid anchors within the assembled PelBC complex, possibly by suppressing their dynamics, and it may facilitate the oligomerization of the subunits into the ring.

The lipid anchors of PelC subunits reside at 3–4 Å distance from the PelB barrel, and so do not interact with the protein. However,

multiple densities were found at the surface of PelB, which occupy the grooves formed by the membrane-facing aromatic and apolar residues (Fig. 5A, B). Those were assigned to acyl chains of the phospholipids, which were either co-purified with the PelBC complex, or were provided upon the nanodisc reconstitution. A fully resolved lipid molecule was located between the PelC subunits H and I, forming extensive interactions with the lipoproteins and the barrel (Fig. 5D). Based on the size and the interactions with the proximate polar residues which coordinate the head group, the density was assigned to a zwitterionic lipid, either PE or PC (Supplementary Figs. 10 and 11). The best-fitting PE could be co-isolated from the bacterial membrane, while PC is a constituent of the synthetic nanodisc, which could replace an endogenous PE molecule upon the reconstitution. Remarkably, a continuous density was resolved in a groove crossing the strands 1–2 and 13–16 that stapled N- and C-terminal ends of the β-barrel. The density spanning from the periplasmic to the extracellular sides of PelB likely arises from averaging of two acyl chains that belong to lipids in the opposing leaflets. In the native bacterial membrane, the groove would be partially filled by a lipid A molecule in the extracellular leaflet. In support of this hypothesis, a cationic cluster of Lys-888, Lys-897, and Arg-921 is found at the extracellular side of the groove, offering a docking site for the phosphate head groups (Fig. 5B). On the other side of the barrel, a similar cluster of Arg-1071 (β-strand 11) and Arg-1102 (β-strand 12) is formed near a hydrophobic groove, which appears to be

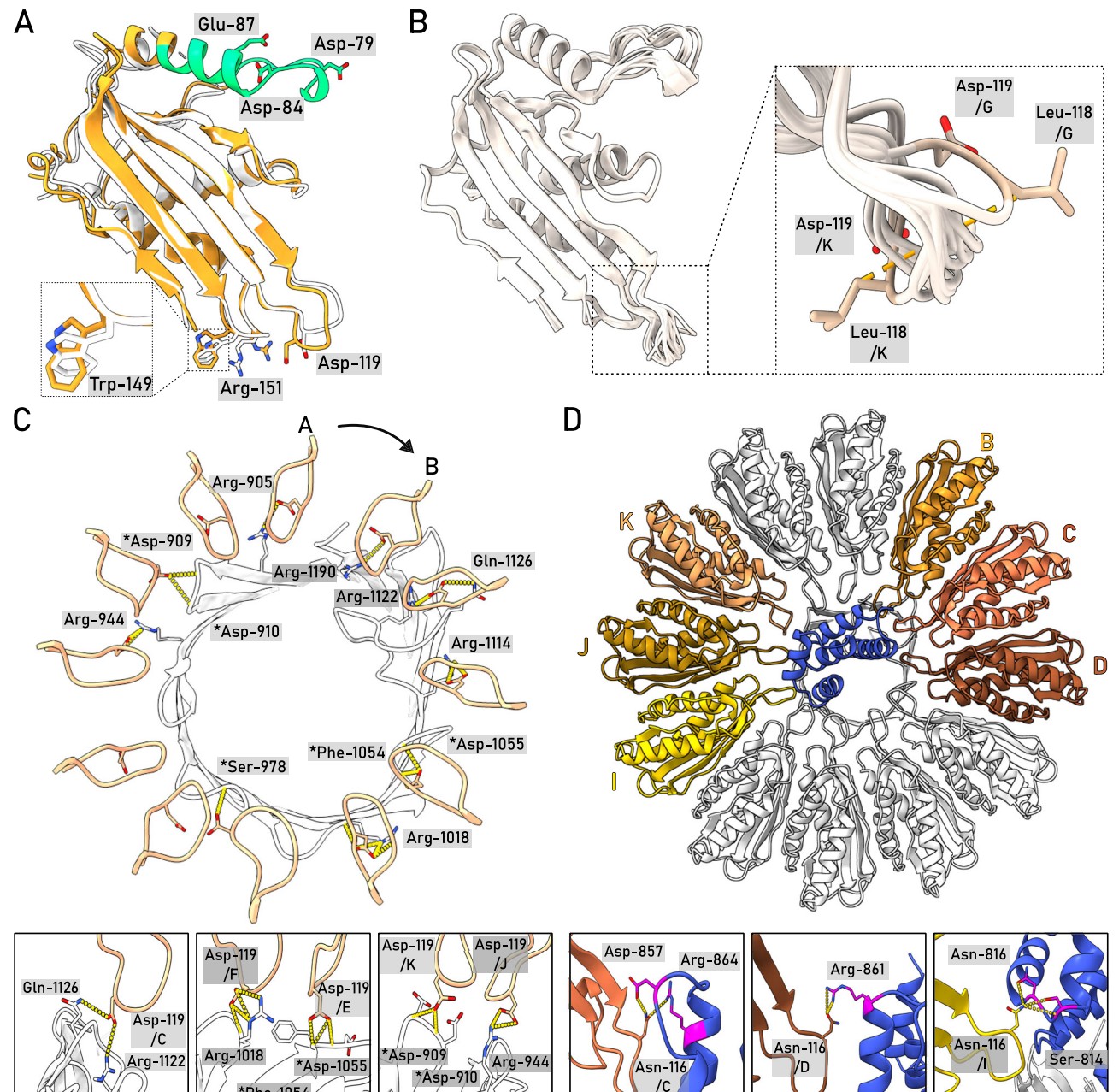

**Fig. 4 | The architecture of the PelC ring and PelC:PelB interactions. A** Cryo-EM structure of *P. aeruginosa* PelC, subunit E (orange), aligned with the partial crystal structure of PelC from *P. phytofirmans* (white). The capping loop absent in the crystal structure is shown in green. **B** Alignment of 12 PelC subunits resolved in the cryo-EM structure. Due to the subunit-specific interactions with PelB, the pore-exposed D-loops manifest different conformations (zoomed), where the deviations measured for Leu-118 Cβ reach up to 8.5 Å for subunits K and G. **C** The residue Asp-119 within the D-loop of multiple PelC subunits is involved in interactions with the periplasmic loops of PelB (residues indicated). PelB residues which interact via the backbone amide group are indicated with asterisk. **D** Several PelC subunits (colored) build electrostatic interactions with the helical scaffold of PelB (shown in blue). Subunits (**C** and **D**) interact with the stalk helix, and subunits B and I-K with the TPR domain.

well-conserved feature among *Pseudomonas* species (Supplementary Fig. 8), suggesting that the PelB structure is evolutionary tuned to facilitate interactions with lipid A molecules.

### The dynamic extracellular loops of PelB open a tunnel for Pel export

The cryo-EM analysis revealed a single conformation of PelB, where the groove within the periplasmic TPR's is aligned with the pore at the extracellular side of PelB between the strands 12–16 and the opposing Plug-I, thus forming a route for the polysaccharide. Notably, Tyr-1103 in the proximity of the pore is conserved among PelB

homologs, being occasionally substituted by phenylalanine, and the residue is also found in the BcsC structure[14], so it may be involved in the EPS translocation, e.g., via CH-pi interactions with the pyranoside rings. However, the exit pore within the resolved PelB structure is occluded by the glycine-rich loop Plug-S (Fig. 3) with a bottleneck width of 4 Å (Fig. 3B). Such a narrow tunnel cannot be employed for the polysaccharide translocation requiring a width of ~8 Å (Ref. 16), so a conformational change, such as dislocation of Plug-S, must occur to open the conduit. As the Plug-O loop is not structurally conserved even among close PelB homologs in *Pseudomonas* species and it extends away from the barrel, it is unlikely that the loop plays a

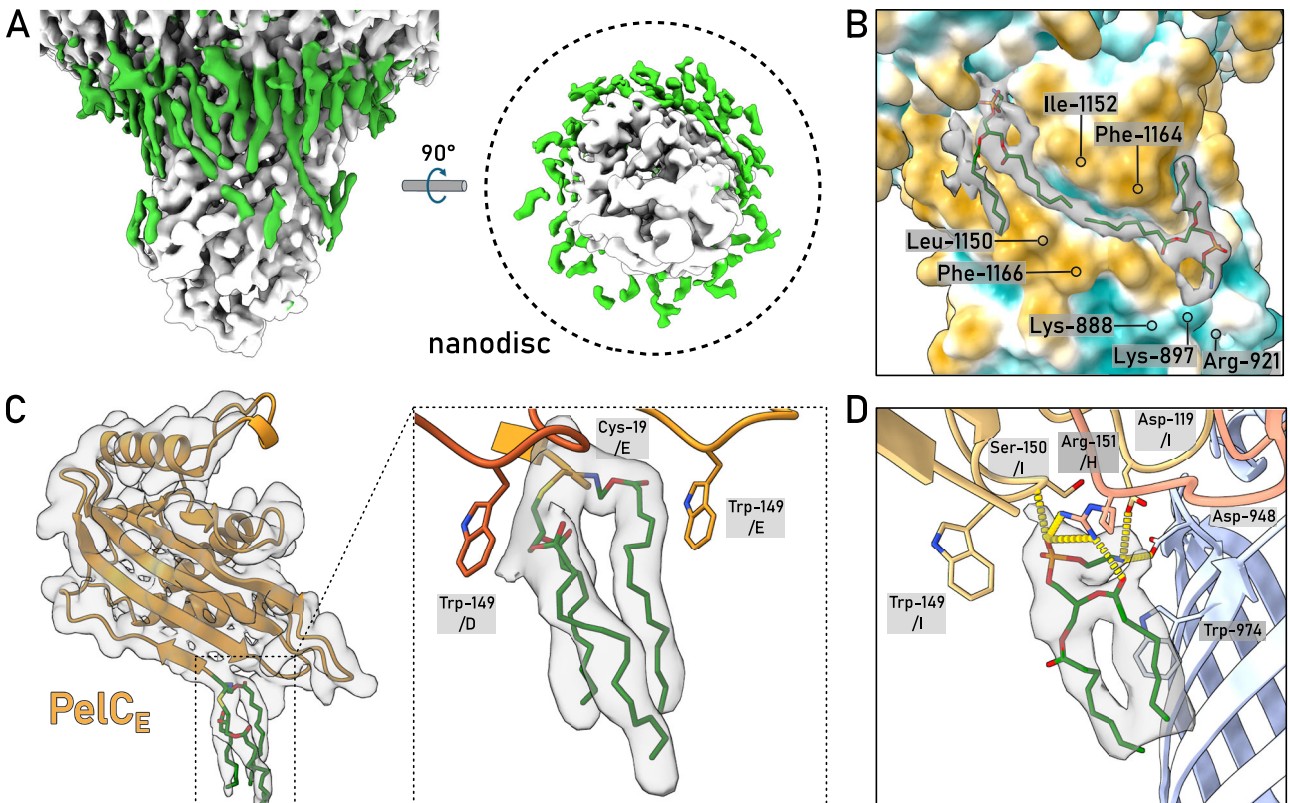

**Fig. 5 | The proximate lipid environment of the PelBC complex. A** Non-proteinaceous densities (green) within the nanodisc assigned to the stably docked lipid acyl chains and the lipoprotein anchors. The approximate nanodisc borders are indicated as a dashed line in the view from the extracellular side (right). The density of the PelC ring is removed for clarity. **B** The acyl chains docked within the hydrophobic groove of the β-barrel formed by Leu-1150, Ile-1152 and mainly Phe-1164 and Phe-1166. At the extracellular side, Lys-888, Lys-897, and Arg-921 form a cationic cluster as docking site for phosphate head groups. **C** The N-terminal acyl chains of the lipoprotein PelC (as example, the subunit E is shown) are stabilized by Trp-149 residues from two subunits, (**D** and **E**). **D** Modelled structure and the interaction network of a phosphatidylethanolamine molecule proximate to PelC subunits H and I.

**Table 1 | Summary of simulations on PelB under different conditions**

| System and conditions | Size [atoms] | Runs | Cumulated time [µs] |
|---|---|---|---|
| EC-LPS, 150 mM KCl, 37 °C | 184,012 | 3 × 500 ns | 1.5 |
| PA-LPS, 150 mM KCl, 37 °C | 145,971 | 3 × 500 ns | 1.5 |
| EC-LPS, 150 mM NaCl, 25 °C | 184,012 | 3 × 500 ns | 1.5 |
| PA-LPS, 150 mM NaCl, 25 °C | 145,971 | 3 × 500 ns | 1.5 |
| POPC:POPG, 150 mM NaCl, 25 °C | 105,713 | 3 × 500 ns | 1.5 |
| POPC:POPG, 150 mM NaCl + 100 mM CaCl$_2$, 25 °C | 105,707 | 3 × 500 ns | 1.5 |
| POPC:POPG, 150 mM NaCl, 25 °C, pH 3.5, 25 °C | 106,029 | 2 × 500 ns | 1.0 |
| Plug-O mutant, POPC/POPG, 150 mM NaCl, 25 °C | 102,598 | 3 × 500 ns | 1.5 |
| Plug-S mutant, POPC/POPG, 150 mM NaCl, 25 °C | 104,249 | 3 × 500 ns | 1.5 |
| Total | | | 13.0 |

*PA P. aeruginosa, EC E. coli, LPS* Lipopolysaccharide, PA-LPS: LPS with one O-antigen unit (outer leaflet) and DOPE:DPPG:DPPE (inner leaflet) at molar ratio 46:31:23[41]. EC-LPS: 100% LPS with five O-antigen units (outer leaflet) and POPE:POPG:PVCL (inner leaflet) at molar ratio 75:20:5.

key role in the protein gating, though its involvement cannot be excluded a priori.

We speculated that the PelB conformation, first of all positions of its extracellular loops, may be affected by multiple physiological factors, such as presence of divalent cations, LPS, and/or variations in the temperature. To test whether those factors induce spontaneous opening of the pore, we established all-atomic molecular dynamics (MD) simulations of the PelB β-barrel (residues 877–1193) in triplicates under a set of relevant conditions. The protein was virtually embedded in a membrane of a tailored composition, and environmental factors could be systematically screened while monitoring the dynamics of the extracellular loops (Supplementary Fig. 12 and Table 1). The experiments were initially performed at 25 °C, a relevant condition for the EPS secretion in surface-based biofilms. First, we examined whether Ca$^{2+}$ cations abundant in the extracellular environment, but omitted in the cryo-EM analysis, could alter the protein dynamics due to interactions with the anionic loops. Equilibration of the nanodisc-like phospholipid-based system in presence of different ions revealed remarkable features at the extracellular side of PelB: The highly charged region consistently attracted cations, and Plug-S formed

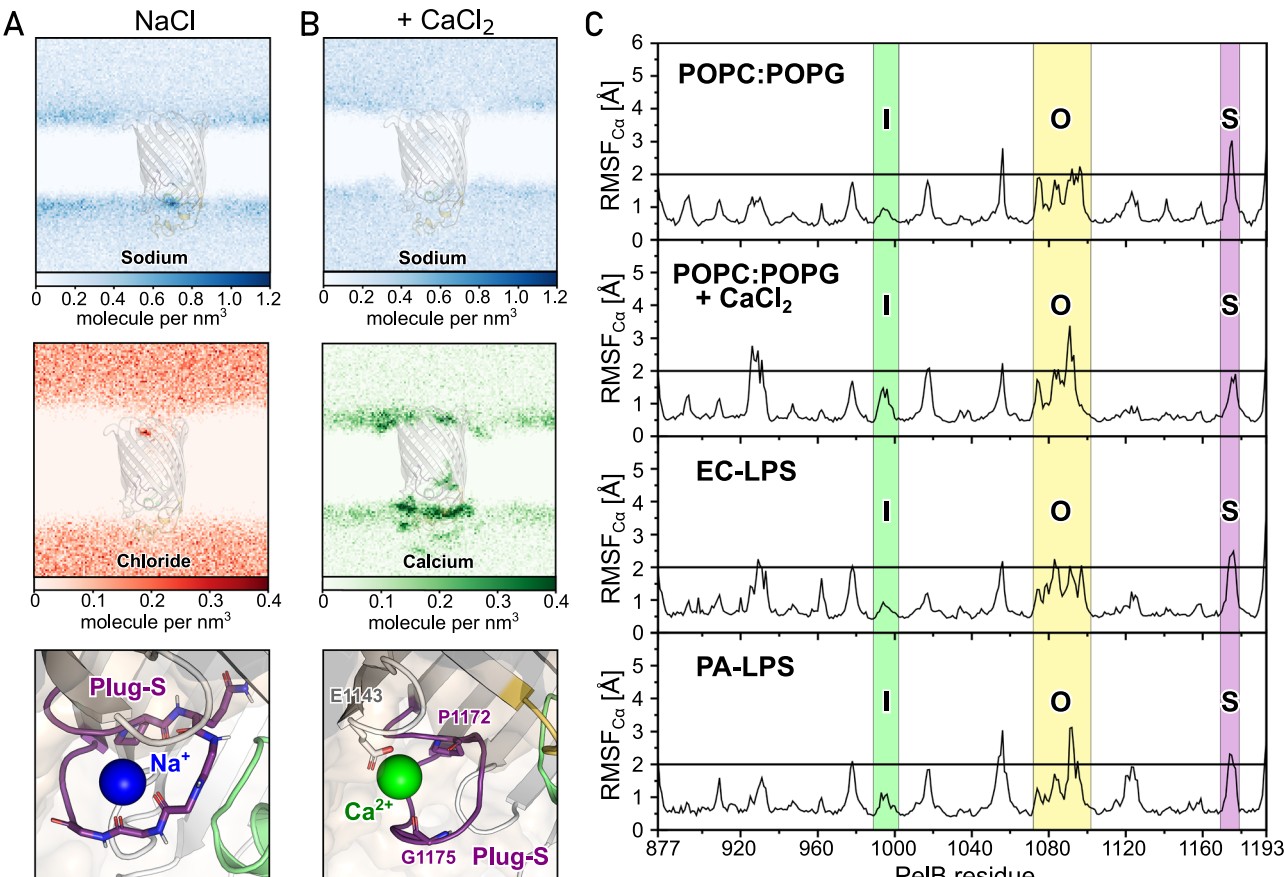

**Fig. 6 | The ion distribution and the PelB β-barrel dynamics in silico. A** The distribution density maps of sodium (top, blue) and chloride (middle, red) ions across the PelB β-barrel embedded into the symmetric POPC:POPG lipid bilayer in 150 mM NaCl. The Plug-S loop stably binds a sodium ion coordinated via the carbonyl groups (bottom panel). **B** The density maps of sodium (top, blue) and calcium (middle, green) ions in the same settings as (**A**), supplemented with 100 mM CaCl₂. A calcium ion binds stably at the side of the Plug-S via the Pro-1172, Gly-1175, and Glu-1143 (bottom panel) and replaces the sodium ion. The density of chloride anions does not change in the presence of calcium. **C** The conformational dynamics of PelB in symmetric POPC:POPG lipid bilayers with and without calcium ions, as well as native-like membranes with an LPS leaflet based on *E. coli* and *P. aeruginosa* models in CHARMM-GUI. The dynamics is depicted via the RMSF of the Cα atoms over the time of 500 ns, and the threshold displacement level of 2 Å is indicated (black line). The regions corresponding to the Plug-I, Plug-O, and Plug-S loops are highlighted in green, yellow, and violet, respectively. Representative simulations of each system are shown. All plots of the triplicates are provided in Supplementary Figs. 13 and 14. Source data are provided as a Source Data file.

stable interactions with a de-solvated sodium ion coordinated by the carbonyl groups of the tetra-glycine loop (Fig. 6A). Once CaCl₂ was included in the simulation, a calcium cation displaced Na⁺ from Plug-S and was stably docked between Pro-1172, Gly-1175, and Glu-1143 from a proximate extracellular loop (Fig. 6B). Despite the significant difference in the ion distribution, the dynamics of PelB were similar between the tested conditions, where Plug-O and Plug-S repeatedly crossed the threshold displacement of 2 Å and so appeared flexible, while Plug-I remained rather static (Fig. 6C and Supplementary Fig. 13).

The model lipid bilayer of the nanodisc does not reflect the natural asymmetry of the bacterial outer membrane, where the extracellular leaflet is composed of lipopolysaccharide (LPS) molecules. Hence, we questioned whether this specific environment enriched with LPS-bound phosphate groups and divalent cations affects the dynamics of PelB extracellular loops. PelB was incorporated into the mimetics of *P. aeruginosa* and *E. coli* outer membranes provided by CHARMM-GUI. The membranes differ substantially by the length of the O-antigen polysaccharides of LPS, which extend into the solvent by 2.5 and 10 nm, respectively (Supplementary Fig. 12). As PelB Plug-O is exposed to the LPS environment, its interactions were analyzed on the example of *E. coli*-type membrane, where we measured the average distance between the Plug-O residues and the specific regions of the LPS (Fig. 7A). The interactions were clustered around three arginines,

i.e., Arg-1071 and Arg-1102 reaching the phosphate groups of the lipid A, and the conserved Arg-1075 interacting with the LPS inner core. As a result of the electrostatic attraction, the former conserved cationic site attracted two LPS molecules (Fig. 7B), and two other LPS molecules were docked by Lys-888, Lys-897, and Arg-921, as predicted from the cryo-EM structure (Fig. 7C). However, no consistent deviations in PelB RMSF values were observed between the simulations in either *E. coli* or *P. aeruginosa* outer membrane models, suggesting that the dynamics of the extracellular loops was not affected by LPS (Fig. 6C and Supplementary Fig. 14).

To test whether the protein dynamics, such as a movement of Plug-O observed in a single trial, are enhanced by the temperature, we repeated the simulations at 37 °C in triplicates. The elevated temperature favored further dislocation of Plug-S, especially for PelB in *P. aeruginosa* membrane (Supplementary Fig. 14). The individual conformations of PelB manifested transient pores, which were sufficiently wide for passage of multiple water molecules, but also for a linear polysaccharide, as exemplified by docking GalNAc-GalN-GalNAc-GalN within the open pore (Supplementary Fig. 15). Thus, the position of Plug-S within the PelB structure together with its high dynamics and the prominent interactions with cations suggest that the loop is the potential gating element for translocation of the cationic polysaccharide.

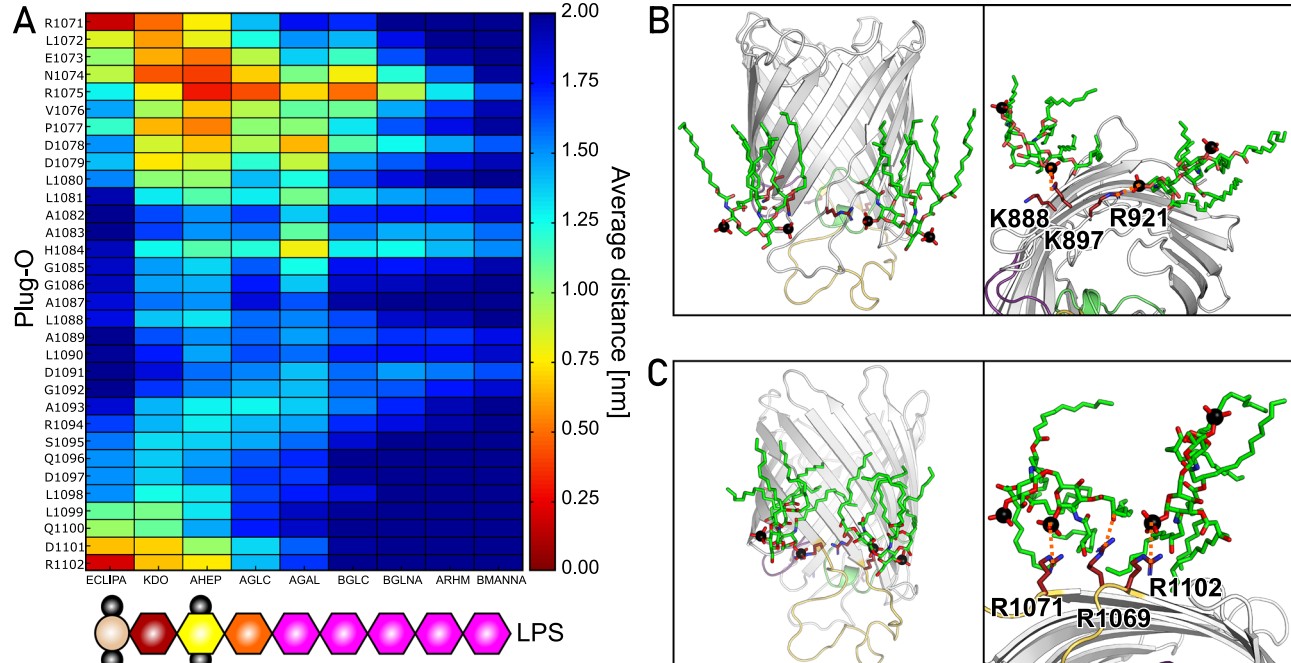

**Fig. 7 | Interactions between the PelB β-barrel and LPS molecules in silico. A** The average distance map between the exposed Plug-O loop and the EC-LPS elements. The schematic structure of the EC-LPS molecule provided by CHARMM-GUI is shown below; details to the abbreviations are provided in the Supplementary Fig. 10E. Here, red to yellow colors imply an interaction between the groups (distances below 1.0 nm), whereas the green to blue colors indicate no direct interactions (distances above 1.0 nm). **B**, **C** Side and extracellular views of two interaction sides of the PelB β-barrel with LPS molecules. The positively charged key residues Lys-888, Lys-897, Arg-921 (**B**) as well as Arg-1071, Arg-1069, and Arg-1102 (**C**) involved in the interactions with the phosphate groups of the lipid A or carbonyl groups of the fatty acids are indicated. Source data are provided as a Source Data file.

## Single-channel conductivity reveals conformational dynamics of PelB

To validate the conformational dynamics of the PelB loops seen in the MD simulations and to document the associated opening of the central pore, we set out to measure the ion conductivity across PelB by means of single-channel electrophysiology. In the electrophysiology set-up, the proteins of interest are embedded into a free-standing lipid bilayer and the ion passage across the membrane is recorded upon applying an electrostatic potential. The conformational dynamics of channel-forming membrane proteins, such as ion channels, toxins, but also outer membrane porins result in the ion flow fluctuations[17], which can be then used to describe the protein dynamics. We assumed that opening of the PelB pore can be monitored in a real-time manner, and contribution of its individual loops can be assessed by testing specific mutants, so the gating element will be identified. As the conductivity experiments require the electrostatic potential applied across the membrane, it could promote dynamics of the charged PelB loops, bringing the protein out of equilibrium. Potentially, the cationic polysaccharide entering the β-barrel of PelB has a similar effect, as it would partially neutralize the charge of the loops and could serve as a trigger for switching the protein conformation.

Similar to the PelBC complex, the PelB protein (residues 762–1193) was expressed in *E. coli* C41(DE3) Δ*acrAB* Δ*ompF* strain, so potential contamination with the major porin OmpF was excluded[11]. To perform the electrophysiology experiment, the detergent-solubilized PelB had to be incorporated into the planar bilayer of 1,2-diphytanoyl-sn-glycero-3-phosphatidylcholine (DPhPC) formed between two chambers of the electrophysiology set-up. The spontaneous incorporation could only be achieved at high ionic strength and pH 3.4, which likely compensated the anionic nature of the extramembrane loops. Upon applying a potential of −150 mV across the membrane, ion currents of various intensities were observed, indicating that the inserted proteins reversibly switched between the conformations of

different conductance (Fig. 8A). Due to stochastic bilayer incorporation and switching between the conformations, ion flow through multiple PelB proteins could be measured, resulting in multiplex opening and closing patterns (Supplementary Fig. 16). Based on the largest common denominating current of ~150 pA, we used only single-channel traces for analysis of conductance and open/closing probabilities (Fig. 8B, C and Supplementary Fig. 17). The recordings were characterized by relatively high noise levels, which originated from thermal fluctuations within the protein structure. As deletion of the periplasmic TPR domain did not affect the recordings (Supplementary Fig. 18), the ion current fluctuations were related to dynamics of the extracellular loops. Notably, the recorded traces showed high consistency for each individual molecule, i.e., the signal fluctuated between certain ion current levels, while the signal levels differed substantially between the measured PelB molecules (Fig. 8B and Supplementary Fig. 17). Thus, the single-molecule detection visualized the otherwise hidden heterogeneity within the PelB ensemble, which could be caused by variations in folding or protein:lipid interactions, e.g., due to co-purified and co-reconstituted *E. coli* lipids and the acquired configuration within the membrane.

To handle the data heterogeneity, we grouped the measured currents into three ranges, i.e., those below 25 pA ("low" conductivity range, L), 25 to 100 pA ("median", M), and above 100 pA ("high", H). Each range must have corresponded to an ensemble of PelB conformations/states, characterized by certain pore dimensions. The individual PelB traces allowed to trace how the ion currents, and so the pore dimensions changed over time, so the occupancy of each conductivity range and the associated currents were calculated for individual molecules and then statistically analyzed based on multiple PelB recordings (Fig. 8C and Supplementary Fig. 19). For the wild-type PelB, the conducting M- and H-states were the most abundant, each constituting of approx. 40%. The ion current levels and the conductance of 0.8 nS observed in the H-range matched the values measured for

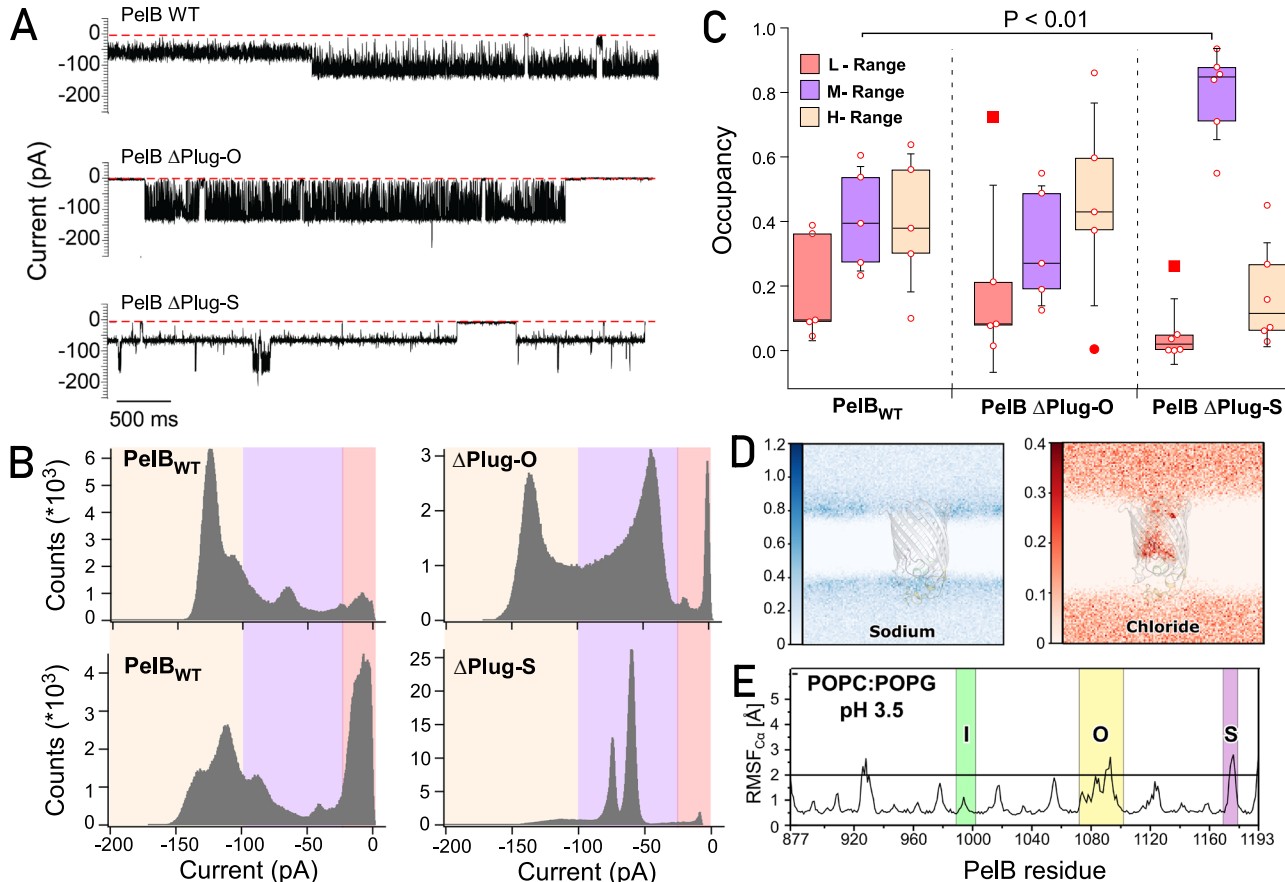

**Fig. 8 | PelB dynamics in single-channel conductivity measurements.**
**A** Recordings of the ion current across individual PelB molecules, either wild-type (WT) or the truncated mutants ΔPlug-O and ΔPlug-S. **B** Representative histograms of the ion current distribution measured for the indicated PelB variants. Colored areas highlight low (up to 25 pA; red), median (25–100 pA; violet) and high (above 100 pA; wheat) conductivity ranges, as described in the text. **C** Occurrence of low (L, red), median (M, violet), and high (H, wheat) conductivity states determined for the PelB variants at the single-molecule level. Each marker indicates a measurement of an individual PelB molecule (n = 5 for PelB_WT and PelB ΔPlug-O, n = 6 for PelB ΔPlug-S). The plots show the median values, the 25th and 75th percentile (boxes)

and the standard deviations (whiskers). The outliers are shown as filled markers (defined by Tukey, beyond 1.5 of interquartile range). The statistically significant difference found for the M-range occupancy is indicated (p = 0.002; one-way ANOVA test). **D** The distribution density maps of sodium (blue) and chloride (red) ions across the PelB β-barrel embedded into the POPC:POPG lipid bilayer at pH 3.5. Chloride anions may enter the interior of the protonated β-barrel. **E** The conformational dynamics of PelB in the POPC:POPG lipid bilayer at pH 3.5. The representation is identical to that in Fig. 6. The regions corresponding to the Plug-I, Plug-O, and Plug-S loops are highlighted in green, yellow, and violet, respectively. Source data are provided as a Source Data file.

OmpG of *E. coli*, where the pore diameter exceeded 1.5 nm[18]. Based on the resolved structure of PelB, we concluded that a major re-arrangement of the sealing loops at the extracellular side would be required to achieve such pore opening, while such conformation was not observed in MD simulations. For the M-state, the conductance of ~0.4 nS matched that of OmpC of *E. coli*, a β-barrel with the pore of ~1 nm[19]. A pore of such dimensions within PelB would be sufficient for translocation of the substrate[16], and the opening could be achieved upon Plug-S dislocation, as suggested by transient conformations seen in MD simulations (Supplementary Fig. 13).

As the conductivity experiment was performed at low pH, we questioned whether PelB retained its conformation and dynamics at these conditions, and so we carried out MD simulations of the lipid-embedded β-barrel while mimicking the solvent acidic environment. The solvent-exposed histidines, glutamate, and aspartates within the β-barrel and at the extracellular surface were protonated at pH 3.5. Differently to the simulations at pH 7.5, we found that Cl⁻ were abundant inside PelB, in agreement with the altered electrostatics, as the repulsion from the anionic residues was abolished (Fig. 8D and Supplementary Fig. 20). Nevertheless, PelB retained its structure along the simulation, and the dynamics of the extracellular loops was barely affected (Fig. 8E). This high stability of the outer membrane protein

may be remarkable, but not surprising: Being exposed to the dynamic environment, the protein must manifest robust stabilization mechanisms, while mild acidification is a common condition met within *P. aeruginosa* biofilms[20].

Single-channel conductivity recordings of the wild-type PelB confirmed the intramolecular dynamics of the protein. As the conductivity fluctuations were likely linked to the movements of the plug domains, we set out to identify their contribution to the gating. Deletion of either Plug-O or Plug-S may affect the channel properties, as suggested by MD simulations and the tunnel calculations (Supplementary Fig. 21). To test the protein dynamics in the conductivity experiments, PelB mutants with the individually deleted/shortened plugs were expressed and isolated (Supplementary Fig. 22A). Among the mutants, deletion of the Plug-I helical region (residues LMRAL) greatly reduced expression and purification yield of the protein, indicating defective folding. The detrimental effect of the Plug-I deletion was further proven when analyzing PelB stability via differential scanning fluorimetry. When measuring the intrinsic fluorescence of the aromatic residues abundant within PelB (15 Trp within the β-barrel), we observed that the wild-type protein, as well as PelB ΔPlug-O and PelB ΔPlug-S underwent cooperative thermal denaturation in the range of 73–75 °C (Supplementary Fig. 22B). In contrast, the construct lacking

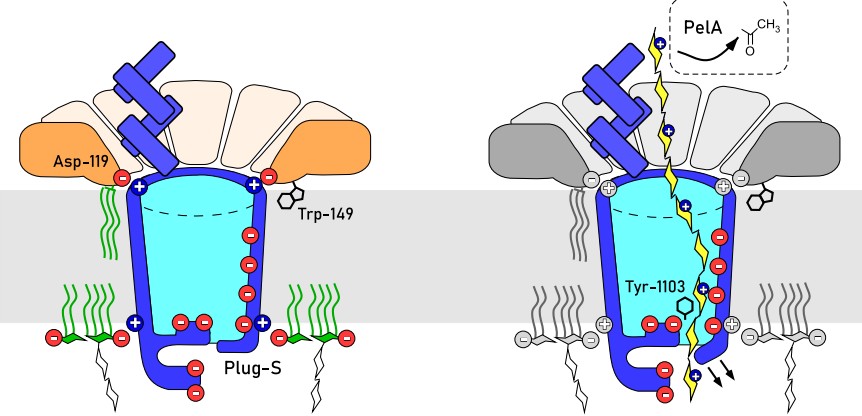

**Fig. 9 | Assembly and the gating mechanism of the exopolysaccharide export complex PelBC.** Left: Schematical overview of the PelBC structure, the charge distribution and the key protein:protein and protein:lipid interactions resolved in the study. Right: The putative route of the substrate, Pel exopolysaccharide (yellow), across the tunneled conformation of the PelBC export complex. The polysaccharide composed of GalNAc is partially de-acetylated by PelA while bound to the helical scaffold of PelB, and the transport is facilitated by electrostatic interactions with the anionic interior of the PelB β-barrel. The polysaccharide uses the exit tunnel formed upon Plug-S displacement, while being screened by the conserved Tyr-1103 residue.

Plug-I unfolded already at 55 °C, in agreement with the stabilizing role of the domain within the β-barrel structure, so the mutant was excluded from the further analysis. Notably, no thermal denaturation was observed for the liposome-embedded wild-type PelB within the experimentally amenable temperature range up to 95 °C, suggesting that the lipid environment offered further stabilization for the protein (Supplementary Fig. 22C). Once the pH level was reduced to 3.5 to mimic the conditions of the single-channel electrophysiology experiments, a transition was observed above 85 °C, suggesting that the protein remained folded within the physiological temperature range, also in the acidic environment.

While retaining the equal stability, PelB ΔPlug-S and PelB ΔPlug-O variants manifested clearly different behaviors in the single-channel conductivity experiments. Deletion of 14 residues within the non-conserved Plug-O loop did not have a significant effect on the observed ion currents and their distribution, as compared to the wild-type protein, so the M- and H-states were predominantly found among the heterogeneous PelB ensemble (Fig. 8 and Supplementary Figs. 19 and 23). In contrast, removal of the Plug-S loop led to radical changes in the conducting properties: Firstly, the recordings manifested substantially lower noise levels, suggesting that those initially arose from high-frequency structural fluctuations of Plug-S. Secondly, PelB ΔPlug-S predominantly showed the conductance within the M-range (occurrence approx. 80%; $P = 0.002$), occasionally switching to H-states (Fig. 8 and Supplementary Fig. 24). The low-conductance L-state was nearly abolished, so deletion of Plug-S rendered a constitutively open pore in PelB. Remarkably, distinct conductivity sub-states were reproducibly observed within the M-range, with the conductance values ranging from 0.42 to 0.57 nS (Supplementary Figs. 24 and 25). Those sub-states must have reflected the dynamics of PelB beyond Plug-S, e.g., movements of the proximate Plug-O loop.

To summarize, the MD simulations and the conductivity measurements jointly suggest that the Plug-S dislocation within the wild-type PelB renders the open state with a pore of approx. 1 nm diameter, which is sufficiently wide for the polysaccharide transport. The dynamics of the gating loop is barely affected by the protein environment, though it is enhanced at elevated temperatures. The high-conductance state observed in the single-channel electrophysiology experiment may be a result of more extensive rearrangements, e.g., displacement/unfolding of the rigid Plug-I or deformation of the barrel under the applied potential. Biophysical and structural analysis in presence of the Pel exopolysaccharide mimetics should provide further insights into the PelBC dynamics and, more generally, the export mechanisms.

## Discussion

Despite the abundance of biofilms in nature and the major roles they play in the microbial pathogenicity, the mechanisms of their assembly at the molecular level remain poorly understood. Resolving the structure and function of the cellular machineries involved in the biofilm formation, first of all exopolysaccharide synthesis and secretion, may assist in developing new strategies to suppress the spreading of many pathogens. Here, we focus on the mechanism of a unique secretion machinery for *P. aeruginosa* Pel exopolysaccharide and present the intact structure of the PelBC complex in the lipid environment, describe the determinants for its assembly, and provide the first experimental insights into the conformational dynamics that may facilitate translocation of the substrate exopolysaccharide (summarized in Fig. 9).

Up to date, crystal structures of several exopolysaccharides transporters have been resolved, which mediate secretion of alginate (AlgE), PNAG (PgaA), and bacterial pEtN-cellulose (BcsC) across the outer membrane[14,15,21]. Consistently, they show β-barrels of either 18 or 16 strands occluded with loops at the extracellular side, and routes for the substrate translocation have been suggested based solely on the structural insights. PelB shares a high structural similarity to BcsC and PgaA, but as a unique feature in Pel secretion, the outer membrane barrel is essentially coupled to the ring of the lipoproteins PelC, which is fully resolved in our structure. The assembly is enabled by the conjugated acyl chains and the membrane-embedded Trp-149 of each PelC subunit, and the necessary electrostatic interactions of multiple lipoproteins with PelB, both with the helical TPR domains and the periplasm-exposed loops (Fig. 9). The structure resolved in the lipid environment and the complementary MD simulations in the native-like membranes described the distribution of lipids and LPS molecules around PelB. While the apolar grooves formed at the outer barrel surface serve for docking the acyl chains, Arg/Lys clusters within the otherwise anionic extracellular interface facilitate local interactions with the phosphate groups of LPS. A number of approaches can be employed for characterizing PelBC:lipid interactions, also in its native membranes of *Pseudomonas* species. Recently, multiple studies high-lighted the potential of amphipathic polymers for extraction of

membrane proteins, followed by the structural and lipidomics analysis[22], though also effects of specific membrane mimetics on a protein structure have been demonstrated[23]. As a complementary tool, computational simulations of the protein:lipid interactions, also in combination with cryo-EM data, allows identifying the interaction hotspots and even the identity of the involved lipids[24]. Beyond the specific protein:lipid interaction, it remains to be shown whether the observed tilt of the PelC ring relatively to the PelB axis has an effect on the membrane morphology, e.g., increased curvature at the periplasmic side, which could be potentially examined when studying PelBC embedded into nanodiscs of larger dimensions[13].

Although resolved in a uniform conformation and in the absence of the substrate, the structure of the PelBC complex provides valuable hints on the polysaccharide transport mechanism. It has been previously suggested that the negative charge at the central pore of PelC is required for electrostatic funneling of the cationic polysaccharide into the membrane channel. However, our data reveal that Asp-119, the key anionic residue within the flexible loop, is required for PelB:PelC docking, as it builds multiple subunit-specific interactions at the interface within the asymmetric complex (Fig. 9). Thus, the polysaccharide transport is rather dependent on the negative potential within the PelB barrel interior, i.e., the charged β-strands 11–14 and extracellular loops. The structural analysis suggests that PelB is a sealed β-barrel, not permeable in its idle state resolved by cryo-EM. Indeed, the outer membrane of *P. aeruginosa* is known to lack porins, which would possibly allow passage of toxic agents across the membrane. Both the MD simulations of PelB in various environments and in vitro analysis of PelB mutants suggest that the opening of a tunnel for the exopolysaccharide secretion is provided by a movement of Plug-S, the extracellular loop aligned with the periplasmic TPR domains and the anionic wall inside the β-barrel. Remarkably, Plug-S is located in a direct proximity to Tyr-1103 from β-strand 12 (Fig. 3C and Supplementary Figs. 7 and 15), a residue which is highly conserved among PelB homologs from different species, including thermophiles, such as *Sulphurihydrogenibium azorense* (gene SULAZ_0918) and *Persephonella marina* (gene PERMA_1356), and also found in the pEtN-cellulose transporter BcsC as Tyr-1030[14]. The residue likely contributes to the channel, where it may coordinate the polysaccharide chain via CH-pi interactions with pyranose rings. Here, further structural and biophysical analysis will be required to reconstruct the path of the Pel chain across the PelB barrel.

The necessity of the PelC ring and the importance of the involved PelC:lipid and PelC:PelB interactions remain puzzling. PelC subunits extensively interact with the TPR domains of PelB, which potentially serve as molecular rails for the Pel exopolysaccharide through the periplasm, resembling the routes followed by polypeptide chains and/or phospholipids in TamB and lipid A in LptA transport proteins[25,26]. In the resolved structure, the groove of the TPR domain is aligned with the pore at the opposite side of PelB, with a broad electronegative surface in between, thus forming a continuous path for the cationic polysaccharide chain. The PelC assembly may assist in stable positioning of the TPR domains above PelB, though other transporters, BcsC and PgaA, equipped with similar TPR scaffolds do not require additional lipoproteins. Alternatively, PelC may facilitate the functionality of PelA, a bi-modal deacetylase/hydrolase enzyme of 105 kDa[7]. PelA interacts with PelB residues 332–436 within the TRP region, where it can access the exopolysaccharide Pel and de-acetylate GalNAc moieties. The dimensions of the full-size PelA make it possible then to reach the PelC ring at the outer membrane, as it is predicted by the modelled structure (Supplementary Fig. 26). As the high processivity of PelA may be required for efficient translocation of Pel, this additional interaction with PelC subunit K would stabilize the PelB-bound enzyme in the functional position. Testing this interaction in vivo and in vitro, also with help of structural analysis, should explain the necessity of this peculiar architecture and result in a comprehensive

model of the exopolysaccharide export at the cellular boundary membrane.

## Methods

### Molecular cloning and protein expression

Gene sequence encoding *P. aeruginosa* PAO1 PelB (PA3063) residues 762–1193 with the conventional N-terminal pectate lyase signal peptide and an octa-histidine tag was synthesized by GenScript (Leiden, Netherlands). The *pelC* gene (PA3062) with the signal peptide of *E. coli* Lpp was synthesized by BioCat GmbH (Heidelberg, Germany). These synthesized genes were subsequently cloned into pETDuet-1 vector, *pelC* in MCS1 (restriction sites NcoI/HindIII) and *pelB* in MCS2 (NdeI/KpnI) (restriction enzymes from New England Biolabs GmbH). For expression of PelB variants, pETDuet-1-based plasmids contained only the respective *pelB* genes. To introduce deletions in loops of the PelB β-barrel, the plasmid was amplified by PCR excluding the target DNA fragment and the template DNA was eliminated upon treatment with DpnI (all used primer sequences are listed in supplementary Table 2). Subsequently, the purified DNA underwent phosphorylation using T4 polynucleotide kinase and ligation employing T4 ligase. Molecular cloning was performed using chemically competent *E. coli* DH5α cells. All enzymes were from New England Biolabs. DNA isolation was performed with PCR & Gel Clean-Up and NucleoSpin Plasmid isolation kits (Macherey-Nagel). The sequences of the recombinant plasmids were confirmed through sequencing (Eurofins Genomics).

*E. coli* C41(DE3) Δ*ompF* Δ*acrAB* cells[11] harbouring the recombinant plasmids were cultured in Luria Bertoni (LB) medium containing 100 μg/mL ampicillin at 37 °C with shaking at 180 rpm. Upon reaching optical density at 600 nm ($OD_{600}$) of 0.6 to 0.8, protein expression was induced by addition of 0.5 mM isopropyl-β-thiogalactopyranoside (IPTG), followed by incubation for 3 h at 30 °C. Cells were harvested by centrifugation at $5000 \times g$ for 15 min at 4 °C. The cell pellets were resuspended in 20 mL resuspension buffer (40 mM HEPES pH 7.4, 150 mM NaCl, 5% glycerol). 0.2 mM AEBSF as well as DNase were added prior to the cell lysis step. Cells were lysed using a shear force (LM20 microfluidizer, Microfluidics), followed by centrifugation at $20,000 \times g$ for 15 min at 4 °C to remove the debris. To isolate the total membrane fraction, the clarified lysate underwent a further centrifugation step at $205,000 \times g$ for 1 h at 4 °C (60Ti rotor, Beckman Coulter). The membrane pellet was then resuspended in the resuspension buffer, shock-frozen and stored at −80 °C until further use.

The localization of the PelBC complex in the outer membrane was verified through ultracentrifugation in sucrose gradient. The total membrane extracts were loaded onto a continuous 20–70% sucrose gradient (w/v) in the resuspension buffer prepared using an automated Gradient Station (BioComp Instruments Ltd.). Subsequently, the gradient was centrifuged at $107,000 \times g$ for 16 h at 4 °C (SW 40 rotor, Beckman Coulter). After centrifugation, the gradient was fractionated from top to bottom using the Gradient Station combined with the UV spectrometer and the fractions were analyzed by SDS-PAGE.

### PelBC isolation and reconstitution into nanodiscs

For isolation of the PelBC complexes and the variants of PelB, the total membranes were solubilized with 2% DDM (Glycon Biochemicals GmbH) in IMAC buffer (50 mM HEPES pH 7.4, 300 mM KCl, 5% glycerol) in 10x volume relative to the membranes. After 90 min incubation, the non-solubilized material was removed by centrifugation at $21,000 \times g$ for 15 min at 4 °C. The soluble fraction was supplemented with 2 mM histidine and loaded onto a gravity-flow column packed with 400 μL pre-equilibrated Ni$^{2+}$-NTA agarose beads (Protino, Macherey-Nagel). Binding was carried out for 90 min at 4 °C with rotation, the flow-through fraction was collected, and the resin was washed four times with 2 mL of the IMAC buffer supplemented with 20 mM histidine and 0.05% DDM. The target protein complex was eluted with the IMAC buffer supplemented 200 mM histidine and

0.05% DDM. Three elution fractions of 800 μL each were pooled, concentrated to 500 μL and injected onto Superdex 200 Increase 10/300 GL size-exclusion chromatography (SEC) column in SEC buffer (50 mM HEPES pH 7.4, 150 mM NaCl, 0.03% DDM) connected to ÄKTA go system (Cytiva). Elution fractions were analysed via SDS-PAGE, and the protein concentration was determined spectrophotometrically based on calculated extinction coefficients of $95,910\,M^{-1} \times cm^{-1}$ for PelB and $497,430\,M^{-1} \times cm^{-1}$ for PelBC assuming 1:12 stoichiometry. Stability of the isolated PelB variants was analysed by differential scanning fluorimetry (nanoDSF) using Prometheus NT.48 instrument (NanoTemper Technologies GmbH). The thermal denaturation was assessed by monitoring the intrinsic fluorescence at 330 and 350 nm between 20 and 95 °C through a temperature ramp of 1.5 °C/min.

Nanodisc-forming protein MSP1D1 was expressed and isolated as described[13]. A lipid mixture composed of POPC:POPG lipids (molar ratio 70:30, Avanti Polar Lipids) was prepared in chloroform, the solvent was evaporated, and the lipids were suspended to the final concentration of 5 mM in 50 mM HEPES pH 7.4 and 150 mM NaCl. The formed liposomes were extruded stepwise through 1 μm and 200 nm track-etch membranes (Whatman) and solubilized using 0.5% DDM for 15 min at 40 °C. Subsequently, the purified PelBC complex was mixed with MSP1D1 and lipids at the molar ratio of 1:4:500 followed by 20 min incubation on ice. After the incubation, pre-washed Bio-Beads SM-2 sorbent (Bio-Rad Laboratories) was added and incubated overnight at 4 °C with rotation. The reconstitution reaction was loaded onto Superdex 200 Increase 10/300 GL column connected to ÄKTA pure (Cytiva), fractionated in 50 mM HEPES pH 7.4 with 150 mM NaCl and analysed via SDS-PAGE. The fractions containing the assembled PelBC complexes in nanodiscs were pooled together, concentrated using Amicon Ultra filters with cut-off of 30 kDa (Millipore) and flash-frozen prior further usage in cryo-EM experiments.

### Cryo-EM sample preparation and data collection

Before plunge-freezing, PelBC nanodiscs at concentration of 0.57 μM (equals to 0.17 mg/mL) were supplemented with (1H, 1H, 2H, 2H-perfluorooctyl)-β-D-maltopyranoside (Anatrace) to a final concentration of 0.03 % to promote random orientation of the particles, and 3.5 μL of the sample were applied to glow-discharged Quantifoil Au 300 mesh R2/2 grids with an additional 3 nm layer of carbon. After incubation for 45 s, the grids were blotted for 3 s and plunge-frozen in liquid ethane using Vitrobot Mark IV (Thermo Fisher Scientific). Data collection was performed at 300 keV using Titan Krios microscope equipped with a Falcon 4i direct electron detector and a SelectrisX Energy Filter (all Thermo Fisher Scientific) at a physical pixel size of 0.727 Å. Dose-fractioned movies were collected in a defocus range from −0.5 to −3.0 μm with a total dose of 60 e- per Å$^2$ and fractionated in 60 frames to obtain a total dose of 1 e- per Å$^2$ per frame.

### Cryo-EM data processing and model building

Gain correction, movie alignment, and summation of movie frames was performed using MotionCor2[27]. Further processing was carried out in cryoSPARC v4.4[28]. A total of 31,712 micrographs were selected and CTF parameters were estimated using PatchCTF. Particle picking was carried out by running a Blob Picker job on a subset of 4040 micrographs. An initial set of 2,345,182 particles were extracted with a box size of 360 pixels (corresponds to 261 Å), binned four times. After 2D classification, selected 2D classes containing 584,172 particles were used in one round of ab-initio reconstruction, followed by multiple rounds of heterogeneous refinement, which resulted in one good class with 386,144 particles. The particles of this class were re-extracted, binned two times, and further refined by non-uniform refinement. Selected templates were used to pick particles on the remaining micrographs of the data set. After multiple rounds of 2D classification a set of 4,773,524 particles was selected for downstream processing. The particles were sorted by heterogeneous refinement, using the refined

volume and the three additional volumes from the ab-initio job. After multiple rounds of heterogeneous refinement, 3,174,321 particles converged to one good class, which was subsequently refined by non-uniform refinement. To further improve the resolution of the PelB β-barrel, masked 3D classification was performed and particles belonging to the classes with best resolved features were combined in a non-uniform refinement job (613,225 particles). Further 3D classification was carried out to improve the N-terminal TPR domain of PelB. A final set of 124,181 particles that converged to the best-resolved class was extracted without binning and refined by a non-uniform refinement job to a final resolution of 2.52 Å.

The complex model began with an AlphaFold2 prediction, which was fitted into the final cryo-EM volume. Subsequent manual adjustments were performed using the COOT program (WinCoot version 0.9.8.92)[29], with ongoing refinement achieved through the option real-space refinement in the PHENIX program (version 1.20.1-4487)[30]. To model the PelC anchor chains, the fatty acid tails (and the glycerol backbone) of the phosphatidylethanolamine structure (PDB: PTY) were extracted and fitted into the density. The subunits of PelC were labelled from A to L, starting with the subunit closest to the first and last β-strand of the PelB barrel.

### Single-channel conductivity recordings

An electrophysiology chamber composed of two 500 μL compartments (*cis* and *trans*) separated by a 20 μm PTFE film with a central aperture of ~100 μm was used for all experiments. To make a lipid bilayer, a drop of hexadecane (4% v/v in pentane) was loaded on the *trans*-side of the PTFE film, and allowed to evaporate for ~2 min. Each compartment was then filled with 400 μL buffer (1 M NaCl, 20 mM citric acid pH 3.4) and ~20 μL DPhPC lipid solution were added (5 g/L in pentane, w/v). Ag/AgCl electrodes were inserted to each compartment: *Trans* was the connecting electrode, *cis* was the ground electrode. By lowering and raising the buffer level in one compartment above the aperture, a lipid bilayer could be formed. The bilayer was equilibrated for 5–10 min before PelB was added. PelB samples were diluted 100–5000× with 15 mM Tris-HCl pH 7.5, 150 mM NaCl, 0.02% DDM and then added to the *cis* chamber in a small volume (below 0.1–0.3 μL). Generally, the protein would insert upon applying the transmembrane potential of −150 mV, while breaking/reforming the bilayer promoted insertion into the membrane. The conductivity data were recorded under a negative applied potential (−150 mV), using an Axopatch 200B patch clamp amplifier connected to a DigiData 1440 A/D converter (Axon Instruments), and using Clampex 10.7 software (Molecular Devices, LLC). Data recordings were made in gap-free setting. Measurements were conducted at 20 °C, 1 M NaCl, 20 mM citric acid pH 3.4 with a 50 kHz sampling frequency and a 10 kHz Bessel filter. Data was digitally filtered with a Gaussian low-pass filter with 2 kHz cut-off prior to analysis. Recordings were analyzed with Clampfit 10.7 software (Molecular Devices, LLC).

In most traces, PelB entered the bilayer in multitude, and the single-channel conductivity was determined from the largest common denominating current. Trace parts with two or more PelB proteins were excluded from the analysis, so only single channels were further examined. Per PelB variant, electrophysiology traces of 8 to 13 s recorded for at least 5 individual molecules were used for analysis. The measured instant currents of individual PelB molecules were plotted as all-points histogram with bin width of 0.5 pA. The regions of low (currents below 25 pA), median (25–100 pA) and high conductivity (above 100 pA) were analyzed individually. The occurrence/occupancy of each conductivity range for individual PelB molecules was calculated by dividing the corresponding count of events over the total counts within the trace and presented as box plots. The outliers were identified according to Tukey's definition, i.e., beyond 1.5 of interquartile range (IQR), while no far outliers were detected (beyond 3 × IQR).

The values were presented as box plots. The average conductance for PelB variants within each range was calculated as:

$$C[\text{nS}] = \frac{I[\text{pA}]}{V[\text{mV}]} \qquad (1)$$

The ion currents and the corresponding conductance values of the sub-states resolved for PelB ΔPlug-S were determined based on Gaussian fits of the individual histograms. Graphs were generated with Prism 9.5.0 (GraphPad Software) and Igor Pro 9 (Wavemetrics).

## Molecular dynamics simulations

All MD simulations reported were performed on HILBERT from the Heinrich Heine University Düsseldorf and SuperMUC-NG at Leibniz Supercomputing Centre in Munich. All systems are summarized in Table 1 and in Supplementary Fig. 12. Two different LPS molecules with its components which are defined as the default for *E. coli* and *P. aeruginosa* were acquired from in CHARMM-GUI (Supplementary Fig. 12D)[31–33]. All atomistic systems were created by CHARMM-GUI and the $Ca^{2+}$ were added in the LPS layer. Furthermore, we used the *mdp* scripts for energy minimization, equilibration in five steps and production run from CHARMM-GUI web server since they are optimized for protein-membrane simulations. The energy was minimized to 1000 kJ $mol^{-1}$ $nm^{-1}$ using the steepest descent algorithm, followed by five-step equilibration to the desired temperature of 298 K (25 °C) or 310 K (37 °C) and pressure of 1 atm to mimic the physiological environment. First, two *NVT* equilibration steps were applied to keep constant the number of atoms (*N*), the box volume (*V*), and temperature (*T*), followed by three-step *NpT* equilibration to adjust the pressure (*p*). The protein's and lipid's heavy atoms were restrained to allow the water molecules and ions to relax around the solute but they were decreased by every equilibration step. The Berendsen thermostat was employed to regulate the temperature in the *NVT* simulations, while the Berendsen thermostat and the semi–isotropic Berendsen barostat were employed for the *NpT* simulations. The PME method was applied to calculate long-range electrostatic interactions with periodic boundary conditions. The van der Waals and Coulombic interaction cutoffs were set to 1.2 nm using the LINCS algorithm to constrain all bond-lengths to hydrogens. Production MD runs were performed for 0.5 μs with a time step of 2 fs by recording the coordinates and velocities every 20 ps as well as the Nosé-Hoover thermostat and the semi–isotropic Parrinello-Rahman barostat.

For all-atomistic MD simulations, the CHARMM36m force field[34,35], modified TIP3P water and GROMACS 2018/2021[36–38] were accomplished while using a pressure of 1 bar. All systems were created by CHARMM-GUI and $Ca^{2+}$ were added in the LPS layer. As protein structure, we used the cryo-EM structure of the PelB β-barrel (residues 877–1193) and the models of the truncated variants, PelB ΔPlug-O and PelB ΔPlug-S[9,39]. When performing the simulations at pH 3.5, the following solvent-exposed histidine, aspartate and glutamate residues were protonated: Asp-895, His-906, Asp-909, Glu-916, His-923, Glu-963, Glu-983, Glu-991, Glu-994, Glu-1047, Asp-1066, Glu-1073, Asp-1079, His-1084, Glu-1141, Glu-1143, and Asp-1149.

To check whether lipids and ions clustered within and around the PelB β-barrel in the in the *xz*-plane, we created density maps of ions and water via *gmx densmap*. Here, the direction to average was over *y*-axis, the grid size was set to 0.08 and the unit was molecule per $nm^3$. We used the DuIvyTools (https://github.com/CharlesHahn/DuIvyTools) to change the format from *xpm* to a matrix *dat* file (*dit xpm2dat*), which could use for plotting the density maps via an own python script.

For describing the stability and flexibility of PelB barrel during the MD simulations, the root-mean-square fluctuations (RMSF) of the Cα atoms around their average positions was calculated for each residue. Similarly to the b-factor, the RMSF value represents the positional change of the selected atoms as time–average, where the residue with a value over 2 Å was defined as flexible. To analyze PelB interactions, we generated average distance maps per residue between **(1)** the extracellular loops, as well as **(2)** to the ions and water, and **(3)** to LPS. We calculated the minimal distance (*gmx mindist*) between the groups by following computing the average distance over time.

To determine the geometry and the dimensions of the tunnel within the PelB β-barrel and its variants, HOLE algorithm (Smart et al., 1996) was employed in combination with the algorithm by MDanalysis (https://docs.mdanalysis.org/1.1.1/documentation_pages/analysis/hole2.html). The tunnels were visualized via VMD. To characterize the overall stability of the PelB barrel in the membrane, we further conducted clustering analyses using the relevant algorithm[40] as implemented in GROMACS. Here, the root-mean-square deviation (RMSD) is a measurement of the Cartesian deviation to a reference structure (mostly the start structure of the MD or a crystal structure). The clustering was employed for all Cα atoms with a RMSD cutoff value of 0.15 nm by using the fitting function before calculating the RMSD on the structure to identify cluster membership. Plotting was performed using Python (v3.10.12) or Gnuplot (v5.4 patchlevel 2).

## Software

Protein structures were visualized, analyzed, and rendered for figures using PyMOL (v2.5.0, Schrödinger, LLC) and ChimeraX (v1.7.1, UCSF). Figures were assembled using Inkscape (v1.3.1) and Canvas X (ACD Systems Inc.). Software used for data analysis or plotting is listed in the respective section above.

## Reporting summary

Further information on research design is available in the Nature Portfolio Reporting Summary linked to this article.

## Data availability

Coordinates and EM map have been deposited at the Protein Data Bank (PDB) under accession code 9H80 and the Electron Microscopy Data Bank (EMDB) under accession code EMD-51916. The MD data have been deposited to the Figshare database (https://figshare.com/s/43b1c784579cb156bdc7). Source data are provided with this paper.

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

## Acknowledgements

We would like to thank Susanne Rieder and Charlotte Ungewickell for the support with cryo-EM sample preparation, and Albert Guskov (University of Groningen) for the initial cryo-EM trials. We thank Birgit Strodel, Florian Altegoer, and Sakshi Khosa (HHU Düsseldorf) and Jens Reiners (Center for Structural Studies, HHU Düsseldorf) for fruitful discussions and commenting on the manuscript. We gratefully acknowledge the computing time granted through the Leibniz Supercomputing Centre (LRZ) of the Bavarian Academy of Sciences on the supercomputer SuperMUC-NG (project pn39gu). Computational infrastructure and support at Heinrich Heine University Düsseldorf were provided by the Centre for Information and Media Technology. The hybrid computer cluster purchased from funding by DFG, project number INST 208/704-1 FUGG, and the Centre for Information and Media Technology (HHU Düsseldorf). The study was supported by the German Research Foundation (Deutsche Forschungsgemeinschaft, DFG; grants Nr. Ke1879/6 to A.K. and Nr. 510674444 to R.B.), the Ministry of Culture and Science of the State of Northern Rhine-Westfalen (project "ACCeSS" to A.K.), the European Research Council (ERC, Advanced Grant Nr. ADG 885711 to R.B.) and the Dutch Research Council (Nederlandse Organisatie voor Wetenschappelijk Onderzoek, NWO; grant Nr. VI.C.192.068 to G.M.).

## Author contributions

M.B. carried out sample preparations and biochemical analysis, and built the structural model based on the cryo-EM data. C.R.H. curated the cryo-EM data collection and carried out the data processing and analysis. S.A.P.S. carried out and analyzed single-channel conductivity experiments. J.L. performed molecular dynamics simulations. O.B. collected and curated the cryo-EM data. R.B. and G.M. curated the data analysis and secured funding. A.K. designed and coordinated the project, curated the data analysis, and secured funding. All the authors wrote and edited the manuscript.

## Funding

## Competing interests

The authors declare no competing interests.
