## [Transparent Peer Review file · Nature Communications]

Assembly and the gating mechanism of the Pel exopolysaccharide export complex PelBC of *Pseudomonas aeruginosa*

Corresponding Author: Professor Alexej Kedrov

Version 0:

Reviewer comments:

Reviewer #1

(Remarks to the Author)

The manuscript 'Assembly and the gating mechanism of the Pel exopolysaccharide export complex PelBC of *Pseudomonas aeruginosa*' by Benedens et al provides novel and interesting insights into polysaccharide export. The relevance of the findings is further increased by the fact that this export is critical for biofilm formation of *P. aeruginosa* which considerably contributes to pathogenicity of this bacterium. The authors present a combination of structural biology, molecular dynamics simulations, and electrophysiology, which allows them to develop a hypothesis for gating for the PelBC polysaccharide export complex. The data appear sound and are presented well.

To further improve and strengthen the manuscript, I only have a few suggestions:

- The authors conclude that a non-annular PE lipids co-purifies with the complex based on modeling of PE in the cryoEM density. This should not be viewed as an absolute proof that the lipid indeed is PE. Even non-annular lipids can exchange with lipids used during reconstitution into nanodiscs. This should be discussed.
- Given the suggested importance of PE, it also would be interesting to see stability data of the complex reconstituted into nanodiscs of different lipid compositions.
- For the representation of PE (SI Figure 9) it would be useful to show additional angles allowing the reader to assess the quality of the fitted model into the density better.
- While the nanodisc is big enough to accommodate PelB, PelC certainly is bigger than the nanodisc. Recently, it has been shown that the size of the nanodisc can have considerable effects on the conformational distribution of reconstituted proteins, even in the absence of obvious protein:nanodisc contacts. This limitation should be discussed in more detail. While not feasible for this study, the use of other membrane systems such as liposomes to solve the structure should be considered and discussed.
- Regarding the conformations of various loops that are deemed important for gating, including the cryoEM densities in the figures would strengthen the arguments that the authors make.
- The MD simulations are difficult to follow for a non-expert. More detailed descriptions of the results would be helpful to increase the understanding.
- Similarly, the electrophysiological experiments deserve and need a better introduction. What exactly is measured, and what exactly can be concluded from this. For example, closed, tunneled, and open are expressions that come out of nowhere. Detailed definitions of what these states mean and how they were assigned are needed.
- I am concerned also about the general assignment of these states from the traces presented. Data were filtered at 1 kHz (methods). Figure 8 does not state the filter frequency, while SI Figure 15 states 2 kHz filtering. Given that many of the events observed are extremely short, I am wondering whether filtering cuts off these events, which in reality could be openings. Presenting analysis or traces at different filter frequencies should be presented to make sure that the different states are not due to artificial data manipulation from filtering.
- Why is the tunneling conductance in SI Figure 15 different? In the extreme case of pore 5, the tunneling state is almost identical to the closed state. Overall, the five pores presented in this figure appear to behave quite differently. Better quantifications and explanations would be beneficial.
- The same is true for SI Figure 18 of PelB deltaPlug-O.
- For SI Figure 19 (deltaPlug-S) how is the closed level defined as virtually no events reach this state. However, there are

many, distinct deviations from tunneled to smaller conductances but these transitions are not assigned to any state.
- The entire electrophysiological data should be carefully revisited and the amplitude histograms should be presented.
- On page 11, line 11, the authors state data not shown. I believe that this is not acceptable anymore. The data should be presented in order for the reader to assess the claims made.

Reviewer #2

(Remarks to the Author)

Benedens et al. report the cryo-EM structure of the PelBC complex from *Pseudomonas aeruginosa* which is localized in the outer membrane of the bacterium and exports the biofilm-forming exopolysaccharide Pel. They describe an almost symmetrical ring of 12 PelC subunits that surrounds the 16-stranded β -barrel of PelB at the periplasmic surface of the outer membrane. It is suggested that the positively charged Pel is driven into the negatively charged interior of the β -barrel, but the pore is blocked by three extracellular loops. The flexibility of one of these loops, revealed by MD simulations, together with electrophysiology of a loop mutant provide the basis for a gating model. However, the functional role of PelC stays enigmatic.

An interesting novel structure is reported with sound experimental data. However, I have some issues with the electrophysiological data. The electrophysiological data were obtained far away from physiological conditions, so the conclusions drawn from it are weakened. The authors report that the insertion of PelB into the membrane requires a pH of 3.4 and high ionic strength. I wondered if the authors tried to change the pH after insertion to 7.5 before measurement to strengthen the result. Furthermore, I wondered at which temperature the electrophysiology was done, as the MD simulations suggested a temperature dependency of gating.

The main conclusion of the e-phys is that the mutation of the Plug-S causes constitutive tunnelling of PelB. No statistic significance is given for this difference in figure 8C. What is designated as closed/tunnelled and open seems to me somewhat arbitrary: For example, if I look at pore 5 in Supp Figure 15, then I see a short signal at 0 pA in the 8 sec trace which I would label as closed, and I don't see any evidence for the tunnelled current as designated by the authors. Thus, it seems to me that the level of the designated "closed" current is indeed the tunnelled level. The histograms and Gaussian fits which were used to determine the currents are not shown.

Furthermore, different currents (beyond the noise) are seen for the tunnelled state. For example, pore 4 in Supp Figure 15, 8s: at the beginning the current is clearly below -50 pA and at the end above -50 pA. This is not discussed at all in the paper. In general, why so different sets of currents are seen for different pores?

Overall, this study is an interesting piece of work that I can recommend for publication in Nature Communications after addressing the raised concerns.

Some minor points:

Fig2, B: the length of the edge or a scale bar should be included.

Fig1. E: also a small band of PelB can be seen at 14 ml elution. Considering the ratio of the band intensities, it might be not too different from the main peak at 10.5 ml

Page 6, line 16: "strikingly" would imply to me that there are more similarities than just a 16-stranded beta-barrel which should have anyway a similar overall structure. Are there more specific similarities not mentioned so far?

Page 6, line 34: is rigid under the exp. conditions.

Page 9, line 1: there should be a reference to Fig 5C

Page 11, line 11: deletion of TPR domain. Is this the complete deletion or as above the partial deletion as used for the EM structure? The data not shown should be shown as SI Figure.

Fig 5, A: The two figures of A show opposite sides of the barrel?

Fig 5, B: (as example subunit E is shown)

Fig 5: B and C label should be swapped to have consistent labelling throughout the figures.

Fig. 6: A and B: The legend should make clearer that A shows a MD simulation in the absence of calcium and B in the presence.

Supp Fig. 9: the three densities shown differ but should be the same? They should also be shown the same way then.

Why were 2 TRPs selected for the construct? The rational or preceding results should be explained.

Page 17, line 33: The concentration of the PelBC-nanodisc sample is not given.

Page 18, line 3 to 5: Å

Page 18, line 4: "0.5 to 3.0 μm " (no minus sign)

Page 18, last paragraph: citations and versions for Coot and Phenix are missing.

Version 1:

Reviewer comments:

Reviewer #1

(Remarks to the Author)

All my original concerns have been adequately addressed.

Reviewer #2

(Remarks to the Author)

The authors addressed all my concerns sufficiently.

Reviewer 1

The manuscript 'Assembly and the gating mechanism of the Pel exopolysaccharide export complex PelBC of *Pseudomonas aeruginosa*' by Benedens et al provides novel and interesting insights into polysaccharide export. The relevance of the findings is further increased by the fact that this export is critical for biofilm formation of *P. aeruginosa* which considerably contributes to pathogenicity of this bacterium. The authors present a combination of structural biology, molecular dynamics simulations, and electrophysiology, which allows them to develop a hypothesis for gating for the PelBC polysaccharide export complex. The data appear sound and are presented well.

Authors: Thank you for the positive evaluation!

Reviewer 1, Comment 1: To further improve and strengthen the manuscript, I only have a few suggestions. The authors conclude that a non-annular PE lipids co-purifies with the complex based on modeling of PE in the cryoEM density. This should not be viewed as an absolute proof that the lipid indeed is PE. Even non-annular lipids can exchange with lipids used during reconstitution into nanodiscs. This should be discussed.

Authors: We admit that the conclusion about the lipid identity cannot be based solely on the cryo-EM data, and this task would require further analysis, e.g. site-directed mutagenesis of the binding pocket in PelB: PelC and lipid-focused mass spectrometry, as well as molecular dynamics simulations of the complete PelBC assembly. Some of those potential directions are discussed in the original manuscript (Page 14, lines 29-37), and we further extended the text, aiming to address the raised issues (Page 8, lines 33-34).

Reviewer 1, Comment 2: Given the suggested importance of PE, it also would be interesting to see stability data of the complex reconstituted into nanodiscs of different lipid compositions.

Authors: We agree that the analysis of protein:lipid interactions on example of PelBC system may be highly informative. However, measuring stability of the complex and interpretation of the results may be not trivial, at least in terms of the thermodynamics: As PelB is highly resistant against the thermal denaturation even when present in the detergent (T_m above 70 °C), and the hydrophobic environment of the membrane further stabilizes proteins (Suppl. Figure 22), it is unlikely to identify the contribution of a single specific lipid docked within the pocket. As shown below, our trials with differential scanning fluorometry on PelB in liposomes (POPC:POPG) suggested that the denaturation temperature lays above 95 °C and so beyond the operational range of the device. Even at acidic pH 3.5 (used in single-channel electrophysiology experiments), the thermal denaturation was observed between 85 and 90 °C.

As an alternative approach in studying specific protein:lipid interactions, we are currently pursuing structural analysis in nanodiscs (please, see the response to the **Comment 4** below), as well as polymer-based isolation of the complex, where native lipids and LPS molecules could be co-purified. Complementary, MD simulations on complex membranes may provide insights about the lipid/LPS distribution around the assembled complex. With those tools, it would be possible to resolve distinct roles of specific lipids and separate them from the bulk properties of the membrane.

Thermal stability of the wild-type PelB in POPC:POPG liposomes tested by nanoDSF. The thermal denaturation is not observed within the amenable temperature range for pH 5.4 and 7.4, and can be only resolved at pH 3.6, occurring above 85 °C. Thus, the membrane environment greatly stabilizes the embedded protein. The data is added to the revised manuscript (Suppl. Figure 22C)

Reviewer 1, Comment 3: For the representation of PE (SI Figure 9) it would be useful to show additional angles allowing the reader to assess the quality of the fitted model into the density better.

Authors: Thank you for raising this point! The figure panels have been modified accordingly.

(continued on next page)

Reviewer 1, Comment 4: While the nanodisc is big enough to accommodate PelB, PelC certainly is bigger than the nanodisc. Recently, it has been shown that the size of the nanodisc can have considerable effects on the conformational distribution of reconstituted proteins, even in the absence of obvious protein:nanodisc contacts. This limitation should be discussed in more detail. While not feasible for this study, the use of other membrane systems such as liposomes to solve the structure should be considered and discussed.

Authors: Thank you for the comment! Indeed, for the experiments presented in the manuscript we chose a rather small nanodisc system based on MSP1D1 scaffold protein. In agreement with the initial assumption, the dimensions of the disc were sufficient to accommodate the b-barrel of PelB, as well as the acyl anchors of PelC lipoproteins. As a follow-up experiment, we recently performed cryo-EM on the PelBC complex reconstituted into a larger nanodisc based on MSP2N2 scaffold lipids (figure below). All the structural features of the PelBC complex, including positions of the resolved lipids and PelC anchors appear highly similar to the initial model (though at lower resolution due to the limited data set). For the manuscript, a discussion of possible experiments in various membrane mimetics is added, according to the Reviewer's suggestion (Page 14, Lines 29-31 and 35-37).

- (A) Cryo-EM reconstruction of PelBC in MSP1D1 nanodiscs (green) overlaid with the reconstruction from a larger MSP2N2 nanodisc (transparent). The actual dimensions of MSP2N2 (above 15 nm) are not resolved due to the averaging.
- (B) Comparison of the lipid densities (green) in the MSP1D1 and MSP2N2 volumes highlight their reproducibility and independence on the nanodisc size. Due to the limited data set acquired for MSP2N2 and the less extensive sorting scheme, higher levels are required to visualize the lipids, thus the lipid “shell” is more pronounced.
- (C) The model of PelBC complex from MSP1D1-based reconstruction fitted into the cryo-EM density from MSP2N2-based nanodiscs.

Reviewer 1, Comment 5: Regarding the conformations of various loops that are seemed important for gating, including the cryoEM densities in the figures would strengthen the arguments that the authors make.

Authors: The requested cryo-EM data was provided in the original manuscript within the Suppl. Figure 4D. To highlight the data, a dedicated Suppl. Figure 5 (also shown below) is introduced in the revised manuscript and multiple views on the relevant regions are provided.

Modelling the structural elements of the PelBC complex

- (A) Zoom-in to the extracellular loops/plug domains of PelB. The fragments of the cryo-EM density map are overlaid with the model of the protein.
- (B) The extracted densities of the plug domains shown in different projections.
- (C) Selection of PelBC structural elements with the corresponding densities from the cryo-EM reconstruction.

Reviewer 1, Comment 6: The MD simulations are difficult to follow for a non-expert. More detailed descriptions of the results would be helpful to increase the understanding.

Authors: Thank you for bringing our attention here. In the revised manuscript, the dedicated chapter was extensively modified/restructured and offered to non-experts for proof-reading. We hope that the resulting text is improved.

Reviewer 1, Comment 7: Similarly, the electrophysiological experiments deserve and need a better introduction. What exactly is measured, and what exactly can be concluded from this. For example, closed, tunneled, and open are expressions that come out of nowhere. Detailed definitions of what these states mean and how they were assigned are needed

Authors: For the revised manuscript, the electrophysiology-oriented chapter has been extensively modified, both in general aspects, such as the introduction, and also the data analysis, as proposed by both Reviewers. We highlight the advantages of the method (single-molecule detection, real-time analysis of protein dynamics, ability to resolve heterogeneity within the sample), but also the limitations (requirement for the transmembrane potential and PelB-specific low pH). The primary contribution of the electrophysiology data was the identification of the Plug-S loop as the flexible region that can be reversibly displaced to open a pore within PelB. The data obtained on the wild-type PelB and the mutants was particularly valuable in combination with the structural insights and the computational modelling of the protein dynamics.

To address the complexity of data in the most general and unbiased way, in the revised manuscript we avoid using the structure-related terms (“closed”, “tunneled”, “open”) and the pre-defined ion current values, which varied between individual molecules (also **Comment 9**). Instead, based on the experiment-derived histograms, we focus on three ranges of ion currents manifested by the measured molecules (below 25 pA, 25-100 pA and above 100 pA), and analyse their occurrence, as well as the average current values within those states on the single-molecule level. Box-whiskers plots are used to present the analysed data, so the scattering between the individual recordings is visualized (Figure 8C and Suppl. Figure 19). In this way, the results obtained on different PelB variants can be easily compared, and the statistical significance of the observed changes can be evaluated. The modifications introduced in the analysis do not affect, but rather strengthen the initially derived conclusions, i.e. the involvement of Plug-S in gating of PelB, and we hope that they allow presenting the complex data in the most adequate and comprehensive way.

(continued on next page)

Reviewer 1, Comment 8: I am concerned also about the general assignment of these states from the traces presented. Data were filtered at 1 kHz (methods). Figure 8 does not state the filter frequency, while SI Figure 15 states 2 kHz filtering. Given that many of the events observed are extremely short, I am wondering whether filtering cuts off these events, which in reality could be openings. Presenting analysis or traces at different filter frequencies should be presented to make sure that the different states are not due to artificial data manipulation from filtering.

Authors: We thank the Reviewer for pointing out these discrepancies. In the revised analysis, we have explored various filtering settings, and found that 2 kHz low-pass Gaussian filter offers an optimal balance between visualizing the otherwise raw data and the pitfall of over-filtering, so it was consistently applied through all the data sets. An example of a conductivity recording subjected to various filtering conditions is shown below (PelB wild-type). As it can be seen from the calculated distribution of the currents (histograms on right side), no pronounced artefacts/smoothing is introduced at the filtering level of 2 kHz in comparison to the unfiltered data.

Filtering exploration of PelB single-channel recordings. **A)** Electrophysiology traces of PelB wild-type with the filtered settings (indicated on the left). Next to each 8 s-trace, a zoom-in of 500 ms is shown. **B)** All-point histograms based on the filtered traces. The data filtered with 2 kHz low-pass Gaussian reflect the features of the unfiltered data set, so it was used throughout the manuscript. Experiments were conducted in 1 M NaCl, 20 mM citric acid pH 3.4. Data was recorded at -150 mV applied potential, 50 kHz sampling rate and 10 kHz Bessel filter.

Reviewer 1, Comment 9: Why is the tunneling conductance in SI Figure 15 different? In the extreme case of pore 5, the tunneling state is almost identical to the closed state. Overall, the five pores presented in this figure appear to behave quite differently. Better quantifications and explanations would be beneficial. The same is true for SI Figure 18 of PelB deltaPlug-O.

Authors: Indeed, substantial heterogeneity was observed between the individual recordings, especially those obtained on the wild-type PelB and the Δ Plug-O variant. At the same time, the ion current values within a single trace showed high consistency, i.e. each individual PelB molecule alternated between several defined conformations. We assume that the heterogeneity within the ensemble of PelB proteins was determined by individual variations in PelB folding or protein:lipid interactions, e.g. due to co-purified and co-reconstituted *E. coli* lipids, as well the acquired configuration within the membrane, i.e. factors that were not possible to control in the employed set-up (discussed on Pages 11-12). To handle this heterogeneity, the new data analysis/visualization approach was taken, as described in the **Comment 7**, and additional channel recordings were added for the Δ Plug-O, so at least 5 pores are analyzed for every PelB variant. Interestingly, the heterogeneity was largely suppressed for PelB Δ Plug-S variant, suggesting that the role of this loop may go beyond the gating of the central pore, but also mediate dynamics/folding of the proximate extracellular loops of PelB.

Reviewer 1, Comment 10: For SI Figure 19 (deltaPlug-S) how is the closed level defined as virtually no events reach this state. However, there are many, distinct deviations from tunneled to smaller conductances but these transitions are not assigned to any state. The entire electrophysiological data should be carefully revisited and the amplitude histograms should be presented.

Authors: Thank you for the suggestions! Selected histograms for each PelB variant have been included to the main manuscript (Figure 8B), and the complete data is presented in the Supplementary Figures 17, 23 and 24, next to the source conductivity traces. The approach was particularly fruitful in application to PelB Δ Plug-S variant, where well-pronounced sub-states were found, as discussed in the manuscript (Page 13, Lines 10-23). We hope that the revised approach to the data analysis and presentation (described in **Comment 7**), as well as additional experimental data (extended data set for PelB Δ Plug-o variant, MD simulations at the acidic conditions) address most (if not all) issues.

Reviewer 1, Comment 11: On page 11, line 11, the authors state data not shown. I believe that this is not acceptable anymore. The data should be presented in order for the reader to assess the claims made.

Authors: We apologize for omitting that! Exemplary recordings are included now as Suppl. Figure 18.

Reviewer #2

Benedens et al. report the cryo-EM structure of the PelBC complex from *Pseudomonas aeruginosa* which is localized in the outer membrane of the bacterium and exports the biofilm-forming exopolysaccharide Pel. They describe an almost symmetrical ring of 12 PelC subunits that surrounds the 16-stranded β -barrel of PelB at the periplasmic surface of the outer membrane. It is suggested that the positively charged Pel is driven into the negatively charged interior of the β -barrel, but the pore is blocked by three extracellular loops. The flexibility of one of these loops, revealed by MD simulations, together with electrophysiology of a loop mutant provide the basis for a gating model. However, the functional role of PelC stays enigmatic. An interesting novel structure is reported with sound experimental data.

Authors: Thank you for the positive evaluation of the manuscript!

Reviewer 2, Comment 1: However, I have some issues with the electrophysiological data. The electrophysiological data were obtained far away from physiological conditions, so the conclusions drawn from it are weakened. The authors report that the insertion of PelB into the membrane requires a pH of 3.4 and high ionic strength. I wondered if the authors tried to change the pH after insertion to 7.5 before measurement to strengthen the result.

Authors: We agree with the Reviewer that the conditions employed in the single-channel conductivity experiments, first of all the acidic pH, drastically differ from those conventionally employed in biochemical/biophysical research. The high ionic strength is a common condition for recording conductance of single channels, so to ensure detectable currents, while the requirement for the acidic pH was specific for PelB. In respect to pH, it should be kept in mind that the exporter PelBC operates in the outer membrane of the bacteria, so the standard pH 7.5 may not represent the actual conditions. Moreover, spontaneous acidification within *P. aeruginosa* biofilms has been shown (doi: 10.1128/AAC.01650-15), while the biofilms grown in mild acidic environment are characterized by the higher density of the EPS-based matrix (doi: 10.1128/spectrum.04832-22).

Following the Reviewer's suggestion, we analysed the currents when changing pH from 3.5 to 7.5 after PelB was incorporated into the membrane. 5-min trace of this experiment is shown below. At pH 3.4, one or two PelB channels fluctuating between different states could be seen. After the pH titration the conductance decreased, so the channels appeared stably closed, in agreement with the uniformly closed structure seen in cryo-EM.

PelB_{WT} inserted in the lipid bilayer in low-pH buffer (20mM citric acid pH 3.4, 1M NaCl). After approx. 2 minutes, pH was increased in a single titration step to pH ~7.5. Single wash step to both *cis* and *trans*, i.e. +100 μ L of 80 mM BTP pH 10.6, 1 M NaCl, mix by pipetting, -100 μ L. Final pH checked with pH paper indicator (Roth).

While the single-channel conductivity experiments at the elevated pH was not informative, we additionally performed all-atomistic MD simulations at pH 3.5 to examine whether the associated protonation (in absence of the potential applied in the electrophysiology trials) influences the protein stability/dynamics (Page 12, Lines 15-26 and Figure 8D and E, Suppl. Figure 20 of the revised manuscript). We observed that the imposed acidic conditions allowed entrance of the chloride anions into the protonated PelB barrel, thus confirming the changes in the charge distribution. However, even at these conditions PelB retained its folded state both within the membrane and at the interface and the dynamics of the plug domains was not substantially affected. Differential scanning fluorimetry (nanoDSF) experiments on PelB in liposomes at pH 3.5 revealed a denaturation signature above 85 C, confirming the high thermodynamic stability of the protein (Suppl. Figure 22C). Thus, we concluded that the low pH alone does not trigger spontaneous opening of the channel, but it should be complemented by further factors, such as the electrostatic potential (*in vitro*) or the charged substrate polysaccharide (*in vivo*). Further experiments with Pel mimetics will be conducted to resolve the substrate-induced gating of PelBC.

Reviewer 2, Comment 2: Furthermore, I wondered at which temperature the electrophysiology was done, as the MD simulations suggested a temperature dependency of gating.

Authors: The experiment was conducted at 20 °C (the information is added to the Methods section, Page 20, Line 17). We did not aim to mimic the elevated temperatures occurring inside the human body, as Pel secretion via PelBC is characteristic to the surface-associated biofilms.

Reviewer 2, Comment 3: The main conclusion of the e-phys is that the mutation of the Plug-S causes constitutive tunnelling of PelB. No statistic significance is given for this difference in figure 8C. What is designated as closed/tunnelled and open seems to me somewhat arbitrary: For example, if I look at pore 5 in Supp Figure 15, then I see a short signal at 0 pA in the 8 sec trace which I would label as closed, and I don't see any evidence for the tunnelled current as designated by the authors. Thus, it seems to me that the level of the designated "closed" current is indeed the tunnelled level. The histograms and Gaussian fits which were used to determine the currents are not shown. Furthermore, different currents (beyond the noise) are seen for the tunnelled state. For example, pore 4 in Supp Figure 15, 8s: at the beginning the current is clearly below -50 pA and at the end above -50 pA. This is not discussed at all in the paper.

Authors: We apologize for rather superficial analysis of the electrophysiology data, also pointed out by the Reviewer 1. For the revised manuscript, the individual recordings have been analysed separately, and the measured current values are presented as histograms (Figure 8A-C and Suppl. Figures 17, 19, 23 and 24). As can be seen from the experiments at the single-molecule level (also recognized in the raw data), the individual PelB recordings manifest substantial heterogeneity in between (see also the **Comment 4** below). While we may only speculate on the origins of the heterogeneity (Page 11, Lines 28-34), we aimed to address the resulting complexity of data in the most general and unbiased way. In the revised manuscript, we avoid using the structure-related terms ("closed", "tunnelled", "open") and the pre-defined ion current values, which varied between individual molecules. Instead, based on the experiment-derived histograms, we focus on three ranges of ion currents manifested by the measured molecules (below 25 pA, 25-100 pA and above 100 pA), and analyse their occurrence, as well as the average currents of those states on the single-molecule level. Box-whiskers plots are used to present the analysed data, so the scattering between the individual recordings is documented (Figure 8C and Suppl. Figure 19). In this way, the results obtained on different PelB variants can be easily compared, and the statistical significance

of the observed changes can be evaluated. The modifications introduced in the analysis do not affect, but rather strengthens the main conclusions, i.e. the involvement of Plug-S in gating of PelB, and we hope that the chosen approach allows presenting the complex data in the most adequate and comprehensive way.

Reviewer 2, Comment 4: In general, why so different sets of currents are seen for different pores?

Authors: Substantial variations in dynamics/stability/functionality of presumably “identical” molecules are a common observation in single-molecule experiments, and also single-particle cryo-EM, where extensive sorting schemes typically reveal the conformational diversity. In the particular case of PelB conductivity, the differences in ion currents and the dynamics may be related to the minor deviations in folding of the protein, occasional presence of co-purified *E. coli* lipids, and variations in the insertion geometry (now described in the manuscript, Page 11 Lines 29-34). Hence, the revised analysis and interpretation of the conductivity data focus on the major observations common for the collected data sets, i.e. (i) the validation of PelB dynamics and (ii) clear effect of Plug-S deletion on PelB channel properties, i.e. predominantly conducting state and severely reduced “noise”, otherwise occurring due to Plug-S flexibility.

Reviewer 2, Comment 5: Fig2, B: the length of the edge or a scale bar should be included.

Authors: Done

Reviewer 2, Comment 6: Fig1. E: also a small band of PelB can be seen at 14 ml elution. Considering the ratio of the band intensities, it might be not too different from the main peak at 10.5 ml

Authors: Thank you for pointing that out. We are confident that the co-appearance of PelB and PelC bands in this case is rather a serendipity. According to the specification of the size-exclusion column used in this experiment (Superdex 200 Increase 300/10), at the elution volume of 14 mL one may expect proteins of 45-75 kDa, e.g. residual monomers of free PelB (the main elution peak at 12-13 mL) and fragments of the PelC ring, while the assembled PelBC complex of 250 kDa (plus the DDM micelle) elutes at the earlier volume of 10.5 mL. The complete SDS-PAGE of this experiment is provided below and it will be also submitted as a Source Data.

Complete SDS-PAGE of the PelBC size-exclusion chromatogram (to Figure 1E)

Reviewer 2, Comment 7: Page 6, line 16: “strikingly” would imply to me that there are more similarities than just a 16-stranded beta-barrel which should have anyway a similar overall structure. Are there more specific similarities not mentioned so far?

Authors: The apparent similarities between PelB and other polysaccharide export channels also include the charge distribution within the barrel, the position of the exit tunnel, presence of a short gating loop and the aromatic residue (Tyr-1103 for PelB) likely involved in routing the substrate. As those features are described at later stages within the manuscript, we adjusted the sentence in question and replaced “strikingly” with “notably”.

Reviewer 2, Comment 8: Page 6, line 34: is rigid under the exp. conditions.

Authors: Thank you, text is corrected.

Reviewer 2, Comment 9: Page 9, line 1: there should be a reference to Fig 5C

Authors: The reference is added.

Reviewer 2, Comment 10: Page 11, line 11: deletion of TPR domain. Is this the complete deletion or as above the partial deletion as used for the EM structure? The data not shown should be shown as SI Figure.

Authors: For this construct, the complete TPR/helical scaffold was removed, leaving only the stalk helix present (identical to the PelB-TPR variant in Figure 1D and E). The single-channel conductivity data is added to the revised manuscript (Suppl. Figure 18)

Reviewer 2, Comment 11: Fig 5, A: The two figures of A show opposite sides of the barrel?

Authors: Correct! In the revised manuscript, the figure is modified to present a side projection and an orthogonal view from the extracellular side (also to highlight the nanodisc dimensions).

Reviewer 2, Comment 12: Fig 5, B: (as example subunit E is shown)

Authors: The text and the figure panel are modified accordingly.

Reviewer 2, Comment 13: Fig 5: B and C label should be swapped to have consistent labelling throughout the figures.

Authors: Thank you for the correction. The references to the figure panels have been adjusted.

Reviewer 2, Comment 14: Fig. 6: A and B: The legend should make clearer that A shows a MD simulation in the absence of calcium and B in the presence.

Authors: In the revised manuscript, the presence/absence of calcium ions is indicated above the ion distribution plots.

Reviewer 2, Comment 15: Supp Fig. 9: the three densities shown differ but should be the same? They should also be shown the same way then.

Authors: Thank you for the suggestion, also in line with the Reviewer 1, Comment 3. The figure was modified accordingly to the comments.

Reviewer 2, Comment 16: Why were 2 TRPs selected for the construct? The rational or preceding results should be explained.

Authors: The studied construct contained three TRP/helical hairpins and the stalk helix (starting from the residue 762, described at Page 5), as we expected their interactions with the PelC ring based on the putative AlphaFold model. However, only two TPR domains were confidently resolved by cryo-EM, suggesting high flexibility of the N-terminal fragment, possibly due to weak interactions with the PelC ring and/or absence of the substrate (Page 6, Lines 17-20). The justification for the construct design was added to the text (Page 5, Lines 10-11)

Reviewer 2, Comment 17: Page 17, line 33: The concentration of the PelBC-nanodisc sample is not given.

Authors: We apologize for omitting this information (0.57 μ M, 0.17 mg/mL), now provided in the Methods section (Page 18, lines 31-36)

Reviewer 2, Comment 18: Page 18, line 3 to 5: Å

Authors: Thank you, corrected.

Reviewer 2, Comment 19: Page 18, line 4: “0.5 to 3.0 μ m” (no minus sign)

Authors: Thank you for bringing our attention to this line. Actually, both values should be negative here, as cryo-EM images are acquired in the under-focus mode.

Reviewer 2, Comment 20: Page 18, last paragraph: citations and versions for Coot and Phenix are missing.

Authors: The information has been added to the revised manuscript.